DOI: 10.1038/s41467-018-02992-9　　**OPEN**

# Warm Arctic episodes linked with increased frequency of extreme winter weather in the United States

Judah Cohen[1,2], Karl Pfeiffer[1] & Jennifer A. Francis[3]

Recent boreal winters have exhibited a large-scale seesaw temperature pattern characterized by an unusually warm Arctic and cold continents. Whether there is any physical link between Arctic variability and Northern Hemisphere (NH) extreme weather is an active area of research. Using a recently developed index of severe winter weather, we show that the occurrence of severe winter weather in the United States is significantly related to anomalies in pan-Arctic geopotential heights and temperatures. As the Arctic transitions from a relatively cold state to a warmer one, the frequency of severe winter weather in mid-latitudes increases through the transition. However, this relationship is strongest in the eastern US and mixed to even opposite along the western US. We also show that during mid-winter to late-winter of recent decades, when the Arctic warming trend is greatest and extends into the upper troposphere and lower stratosphere, severe winter weather—including both cold spells and heavy snows—became more frequent in the eastern United States.

[1] Atmospheric and Environmental Research, Lexington, MA 02421, USA. [2] Department of Civil and Environmental Engineering, Massachusetts Institute of Technology, Cambridge, MA 02139, USA. [3] Department of Marine and Coastal Sciences, Rutgers University, New Brunswick, NJ 08901, USA. Correspondence and requests for materials should be addressed to J.C. (email: jcohen@aer.com)

Variability in the day-to-day weather is due to a combination of forced and natural variability. Forced variability results from boundary conditions, such as sea-surface temperatures, and natural or internal variability results from the chaotic nature of dynamical systems[1,2]. While the tropics are usually considered the main driver of boundary-forced variability[3,4], recent studies have argued that the Arctic is playing an increasingly important role as a boundary-forcing agent owing to its accelerated warming relative to other regions of the globe[5–9].

Increasing greenhouse gases are contributing to a general warming of the atmosphere and oceans globally[10]. Over recent decades, warming has dominated global temperature trends during three of the seasons[11]. In winter, however, cooling trends have been observed across Eurasia and the eastern US[12–14] along with rapid warming in the Arctic[5,6,15]. This seesaw winter temperature pattern is known as the "warm-Arctic/cold-continents pattern"[16]. A vigorous debate in the climate community is whether and/or how much the Arctic can influence mid-latitude weather[9,17] and, in particular, whether a warmer Arctic increases the likelihood of severe cold spells in the mid-latitude continents[18].

Anthropogenic global warming is widely expected to increase certain types of weather extremes, including more intense and frequent heat waves and droughts as well as heavy precipitation events[19–21]. Surprisingly, however, over the past two to three decades, the increase in extreme weather has included more (not fewer) severe cold-air outbreaks and heavy snowfalls observed both in North America and Eurasia[6,12,15,18,22–25].

Previous studies have shown qualitatively that anomalously high geopotential heights across the Arctic are linked with extreme weather events across the mid-latitudes in winter[18,26] and even into spring[27]. However, those studies were limited to just a few months of one particular year. Here we present a more extensive, quantitative analysis of the link between Arctic variability and severe winter weather across the mid-latitudes. In this study we find a robust relationship between Arctic temperatures and severe winter weather in the United States. When the Arctic is warm both cold temperatures and heavy snowfall are more frequent compared to when the Arctic is cold. We also found that during the period of accelerated warming when the Arctic warming reaches into the upper troposphere and lower stratosphere during mid-winter to late-winter severe winter weather has been increasing.

## Results

**Metrics analyzed**. We employ three metrics to diagnose the relationship between Arctic temperatures and severe winter weather. The first two are called the polar cap geopotential height anomaly (PCH) index and the polar cap air temperature anomaly (PCT) index. The PCH and PCT indices measure the area-averaged geopotential height and temperature anomalies poleward of 65° N and from 1000 to 10 hPa. Both PCH (units in meters) and temperature (units in °C) are normalized by their standard deviation. The PCH values incorporate air temperature and surface pressure, thus combining both thermodynamic and dynamic influences[28]. PCT reflects only thermodynamic effects.

The third metric is the Accumulated Winter Season Severity Index (AWSSI)[29]. We analyzed changes in daily and cumulative AWSSI in relation to changes in PCH/PCT at diverse geographic locations in order to explore the relationship between Arctic variability and severe winter weather (Methods section). The AWSSI diagnoses severe weather owing to extreme snowfall and temperatures at individual stations across the US. It is reported as an accumulated value throughout the winter season, which informs comparisons of weather severity between years. Daily accumulated changes in AWSSI provide insight into episodic severe winter weather. For our study, the AWSSI is advantageous because it integrates both intensity and duration of temperature, snowfall, and snowcover into one index to measure weather severity across seasons and stations. However, the thresholds used to create the index are somewhat subjective. The AWSSI index is incremented based on thresholds of maximum and minimum temperature, snowfall and snow depth. Because the AWSSI index is not increased unless temperatures fall below freezing and snowfall or snowcover exists, the index better represents winter weather variability in cities that experience colder temperatures and/or heavy snowfall, such as those in the mid-west relative to those in the southern US or the west coast.

**Arctic variability and mid-latitude weather**. The daily change in seasonal AWSSI (or the daily accumulation for that day) is composited for all standardized PCHs at 12 representative cities across the US (see Fig. 1 for the geographic distribution of the chosen stations) by computing the mean change in AWSSI associated with daily PCH values at each isobaric level during winter (DJF) from 1950 to 2016 (Fig. 2). A strong relationship between a warmer Arctic and increased frequency of severe

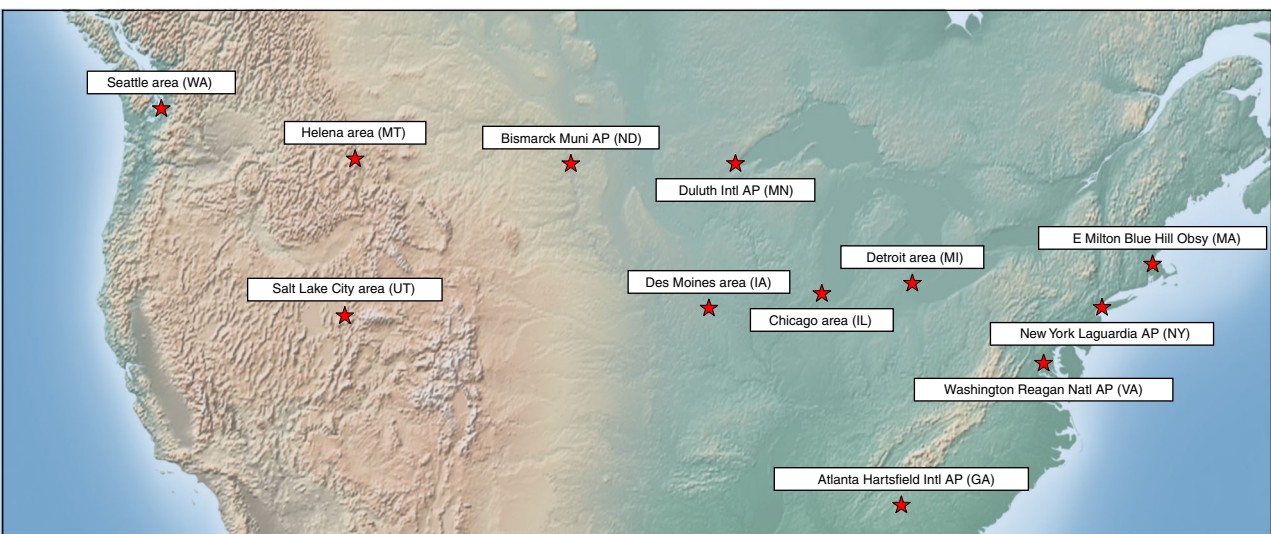

**Fig. 1** Geographic locations of cites analyzed. We chose a geographically diverse set of 12 cities to analyze Arctic variability and severe winter weather, though we chose more cities in the northeastern and mid-western US where severe winter weather is more common than in other regions of the US

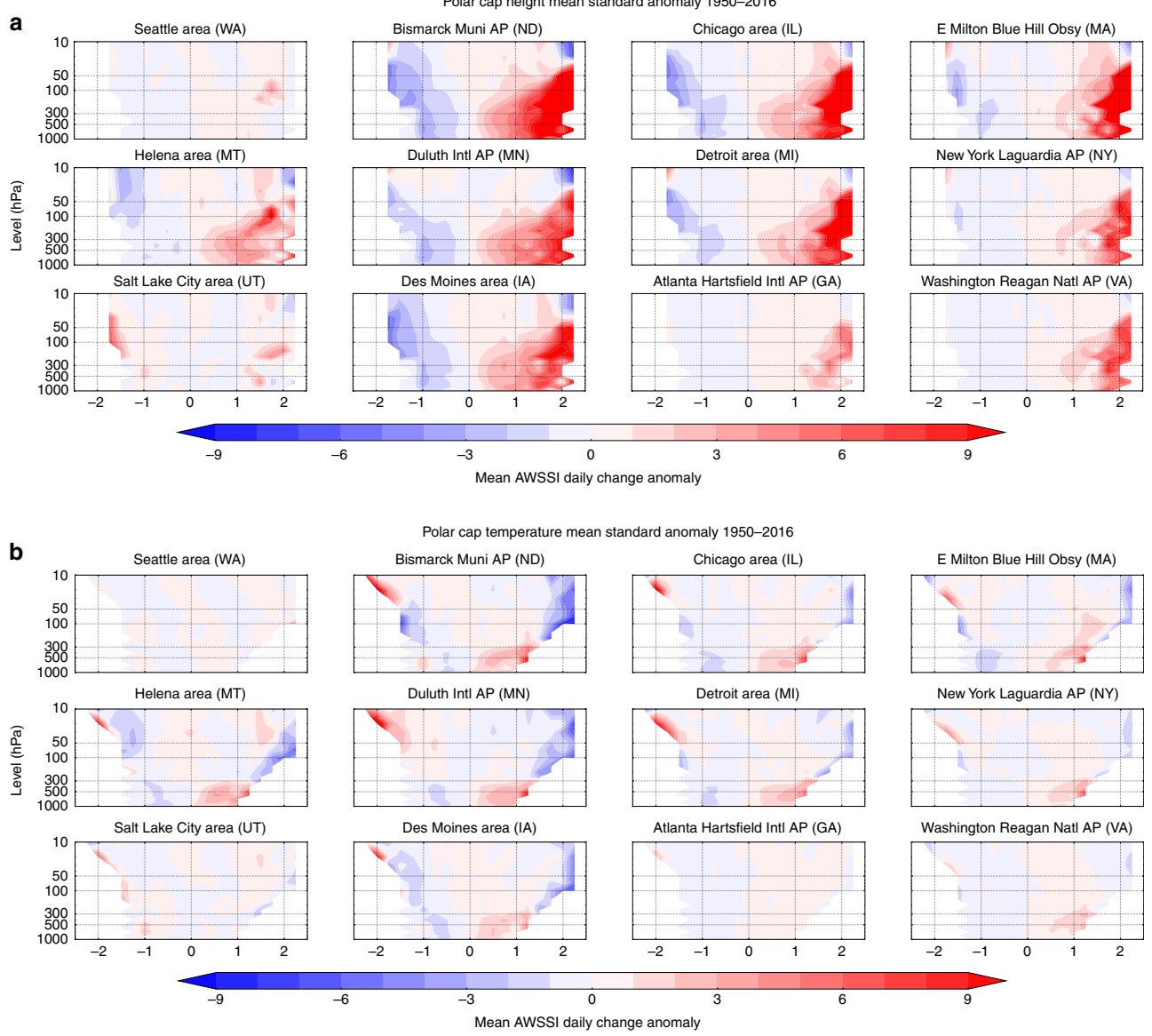

**Fig. 2** Warm Arctic related to increased severe winter weather. The departure from the winter average in daily change in the AWSSI (Accumulated Winter Season Severity Index) at several weather stations across the US during December–February shown at all levels between 1000 and 10 hPa. AWSSI is plotted with composited values of the polar cap geopotential height (PCH, **a**) and air temperature (PCT, **b**) standardized anomalies from the surface to the mid-stratosphere (10 hPa), north of 65° N, from 1950 to 2016. Anomalies computed relative to climatology from 1981–2010. Results for all stations are consistently statistically significant at $p < 0.01$. Statistical significance for both PCH and PCT at selected levels for all cities shown in Supplementary Table 1

winter weather is apparent for all stations east of the Rockies, with the strongest association in the eastern third of the US, where we find a statistically significant ($p < 0.01$) and nearly linear relationship between Arctic height changes throughout the troposphere and AWSSI. When Arctic heights are at their lowest (PCH < ~ −1), severe winter weather is unlikely. For larger values of PCH (PCH > +1), the likelihood of severe weather increases, with correlations peaking when the PCH is greater than +1.5. This relationship is fairly consistent throughout the troposphere over the full range of Arctic height anomalies. The correlation generally holds in the stratosphere (below the 30 hPa surface) as well. In the Rockies and along the west coast, however, the relationship is weak, and some stations even exhibit the opposite relationship, i.e., a relatively warm Arctic favors milder winter weather. This result is consistent with the predominance of an anomalous western ridge during the recent period of pronounced Arctic warming.

We also investigate the separate association between PCH variability and extreme cold temperatures or heavy snowfall (Figs. 3 and 4). Positive values of PCH exhibit a stronger and more extensive relationship with temperature than with snowfall. The most robust relationship between PCH and snowfall is in the northeastern US, thus it is likely that snowfall in this region is most sensitive to Arctic variability, with higher Arctic geopotential heights and relatively warmer temperatures favoring heavier snowfalls[30].

As discussed earlier, the PCH combines both thermodynamic and dynamic influences. To determine whether a direct relationship exists between Arctic temperatures alone and mid-latitude weather, we have repeated the prior analysis substituting PCT for PCH (Fig. 2), demonstrating that a significant fraction of the relationship between Arctic variability and severe winter weather across the mid-latitudes is related to variability in Arctic tropospheric temperature.

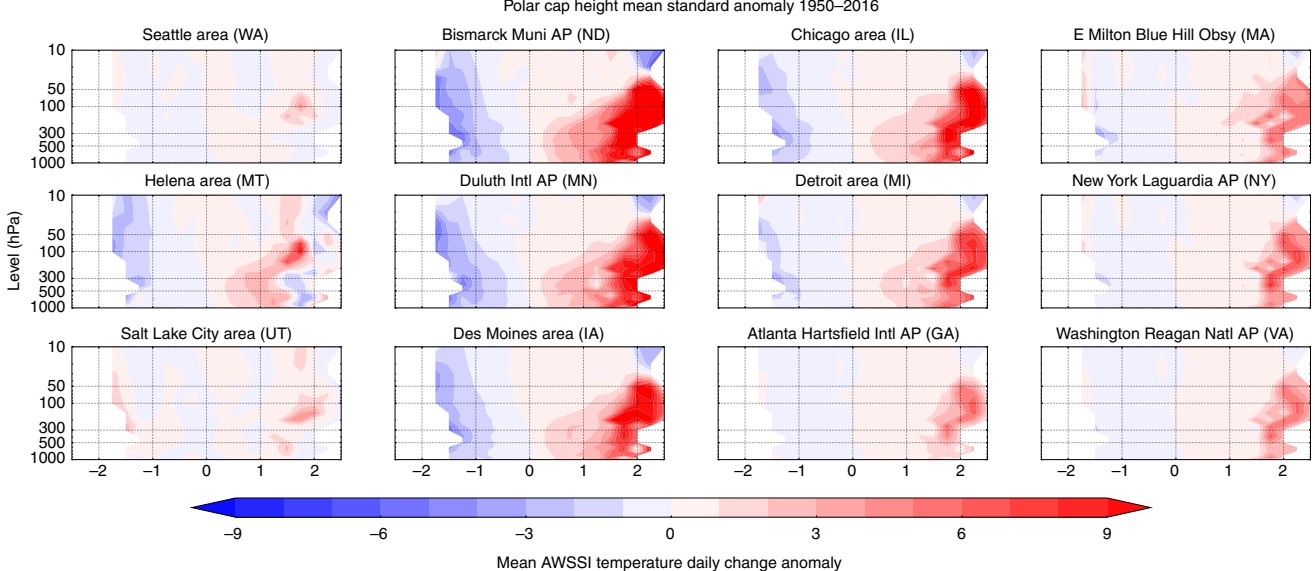

**Fig. 3** Warm Arctic related to colder winter temperatures. Temperature contribution to the average daily change in the AWSSI (Accumulated Winter Season Severity Index) at selected weather stations across the US associated with polar cap geopotential height anomalies (PCH) from the surface to the mid-stratosphere (10 hPa) 1950–2016. Anomalies computed relative to climatology from 1981 to 2010

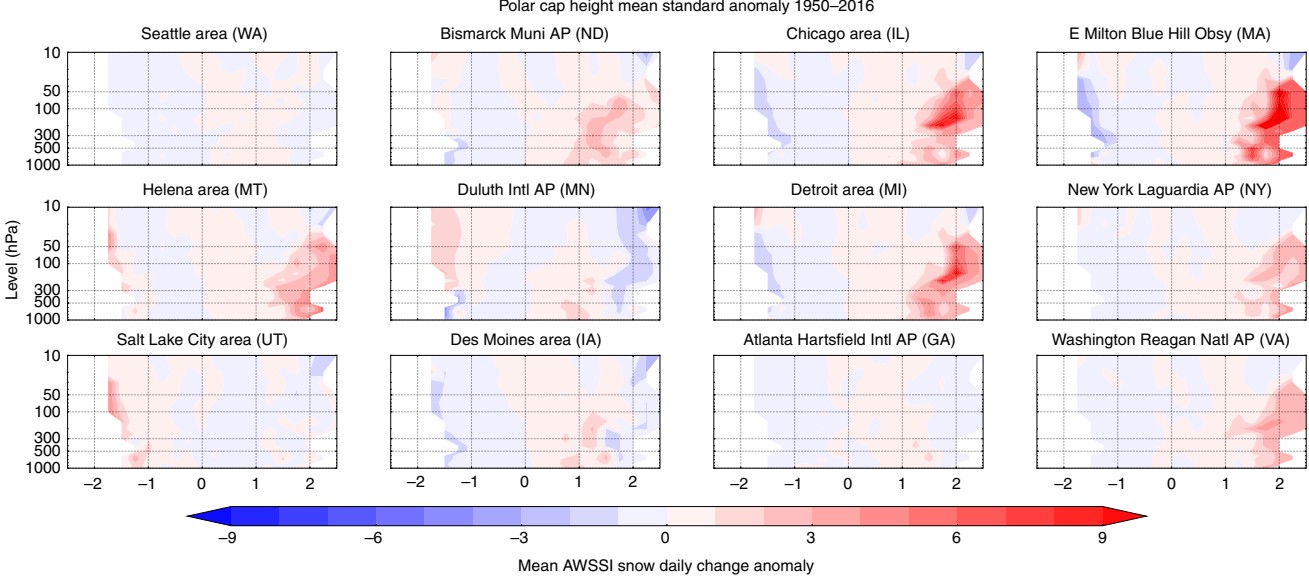

**Fig. 4** Warm Arctic related to increased snowfall. Snowfall contribution to the average daily change in the AWSSI at selected weather stations across the US associated with polar cap geopotential height anomalies (PCH) from the surface to the mid-stratosphere (10 hPa) 1950–2016. Anomalies computed relative to climatology from 1981 to 2010

The relationship between PCH and AWSSI is stronger than the relationship between PCT and AWSSI. Given that PCT represents only thermodynamic influences, it is not surprising that PCH is more strongly correlated with severe winter weather than is PCT. As we show in Supplementary Tables 1 and 2, however, the statistical significance of the correlations between AWSSI and either PCH or PCT are extremely high throughout the troposphere. A notable distinction is in the stratosphere, where PCH/AWSSI correlations are strong and PCT/AWSSI correlations are weak, consistent with another recent study[27]. This finding suggests that the tropospheric Arctic warming contributes most to higher PCH in the stratosphere, which is associated with increased severe winter weather in the eastern US. Previous work suggested that a weakened stratospheric polar vortex (SPV) is related to colder temperatures across mid-latitude continents, including the eastern US[25,26,31–33]. Our analysis refines this notion, suggesting that high geopotential surfaces in the Arctic stratosphere are more important than warm stratospheric temperatures in forcing mid-latitude severe winter weather. Further analysis is needed to confirm this relationship.

Because this is an observational study, cause and effect cannot be determined, and it is possible that the conditions favoring severe winter weather and amplified flow (or larger Rossby waves) also favor Arctic warming. To address this issue, we calculated the lag correlation between PCH at 500 hPa and AWSSI from −30 to +30 days (Fig. 5). We find that correlations peak when the PCH leads AWSSI by five days, then quickly decrease after zero lag and even become slightly negative when the AWSSI leads. These

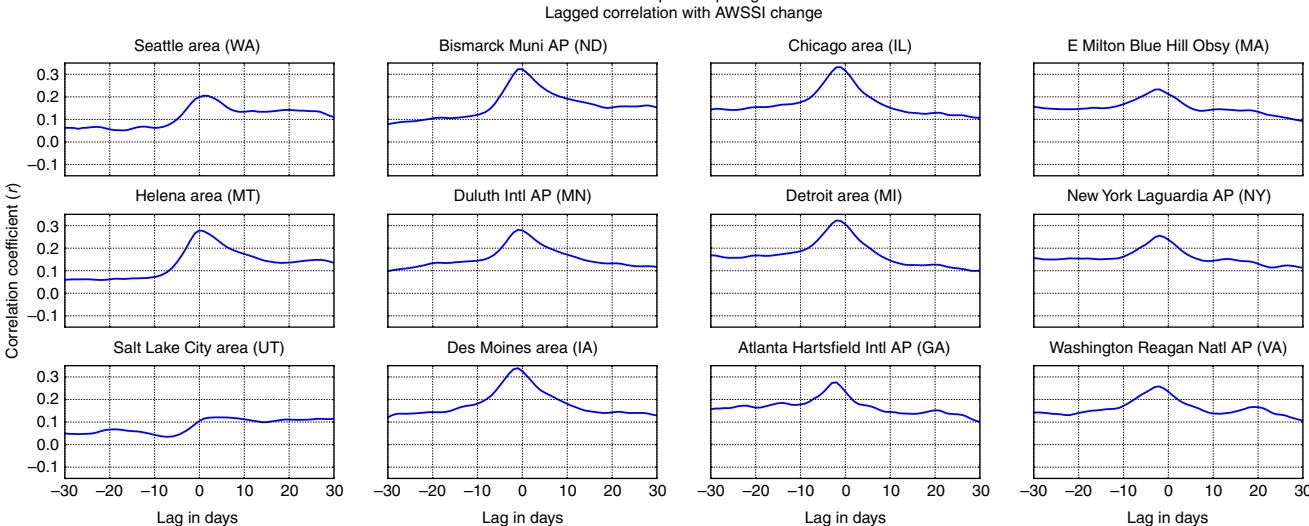

**Fig. 5** Arctic variability leads occurrence of severe winter weather. The correlation between average daily change in the AWSSI at selected weather stations across the US associated with polar cap geopotential height anomalies (PCH) for all days between ±30 days. The peak value is reached when the PCH leads the AWSSI by 5 days

results imply that positive PCHs precede the occurrence of severe winter weather, rather than circulation patterns associated with severe winter weather being the main driver of positive PCH anomalies.

We further analyzed AWSSI variability during the 2-week period 5 to 19 days following high PCH values. We find that severe winter weather is more likely in the eastern US during multiple weeks after positive PCH values occur in the upper troposphere and lower stratosphere (Fig. 6). One interesting difference between the contemporaneous PCH/AWSSI versus the PCH values that lead AWSSI by 2 weeks or more is that significant correlations extend well into the mid-stratosphere when PCH leads. This is consistent with previous results showing that the occurrence of severe winter weather is more likely over multiple weeks following a weak SPV[24,25], demonstrating the potential for using Arctic variability to predict the likelihood of extreme winter weather with leads greater than the synoptic time scale (on the order of days).

To compare Arctic versus tropical influences on severe winter weather events, the analysis was repeated but with the PCH index replaced with the El Niño/Southern Oscillation (ENSO) index, as the tropics are generally thought to be the most important remote driver of mid-latitude weather[34]. In Supplementary Figure 1 we plot the composite AWSSI relative to the standardized Niño 3.4. For all stations across the country, there is no preferential value of AWSSI with ENSO variability, though there does seem to be a decline in severe winter weather for the most extreme El Niño values. This finding suggests that Arctic variability has a stronger influence on severe winter weather events than does ENSO variability.

The AWSSI is calculated only for the US, but to provide a sense of applicability of this approach to the whole NH, we composited hemispheric surface temperature anomalies based on cold [−3.0 to −0.5] and warm [0.5 to 3.0] PCH and PCT values (Fig. 7). In the transition from a relatively cold to warm Arctic, the mid-latitude continents transition from warm to cold temperature anomalies. This relationship is especially apparent in central and southeastern North America, northern Europe, northern Asia, and East Asia. The relationships between PCH/PCT and AWSSI exhibited in the US, therefore, appear to be valid across northern Eurasia, consistent with other recent studies[35,36].

**Arctic mid-latitude linkages in era of Arctic amplification**. It is well documented that the Arctic is warming at a rate two to three times faster than the global average, a phenomenon known as Arctic amplification (AA)[37–39]. While AA is anticipated to reduce the severity of cold-air outbreaks and heavy snowfalls[40–42], cooling trends have dominated NH continents since the emergence of rapid Arctic warming around 1990[6,15,23], contrary to expectations.

One reason offered for this counter-intuitive cooling is the two decades cooling trend in tropical Pacific sea-surface temperatures that resemble a La Niña pattern[4]. An alternative explanation is that AA is modifying the large-scale circulation where winter cooling over NH continents is favored[5,8,13,15,24,25,33]. Furthermore, recent climate trends—including mid-latitude temperatures—better match Arctic warming trends than tropical cooling trends[4,43]. This hypothesis is controversial, however, owing to large internal variability and because pre-AA-era studies of mid-latitude dynamics identify the tropics as the predominant driver of change[9,17]. Furthermore, global climate model simulations that correctly reproduce AA indicate that cold extremes and seasonal snowfall will continue to decrease as the globe warms[40–42]. Challenges to the idea of Arctic influence are discussed in greater detail below.

In Fig. 8 we plot the trends in daily PCH and PCT throughout the winter from 1990 to 2016. The tendencies are consistent with the observed warming trend in the Arctic and are not limited to the near-surface, but rather extend throughout the troposphere. The stratosphere cools in early winter but warms in mid-winter, consistent with a reported increasing frequency of sudden stratospheric warmings (SSWs)[24]. Also shown in Fig. 8 are daily trends in AWSSI for selected cities. Despite model projections for decreasing cold extremes as global warming intensifies, the trends in AWSSI during recent decades are more complex. In the western US, where no strong relationship between Arctic warming and AWSSI is apparent (Fig. 2), severe winter weather has generally decreased since 1990. In contrast, in the eastern US during periods when PCH trends are largest and during periods of substantial stratospheric and upper tropospheric warming (15 January–15 February), severe winter weather has increased. This is consistent with the result that severe winter weather is more likely for multiple weeks following SSW events (Fig. 6).

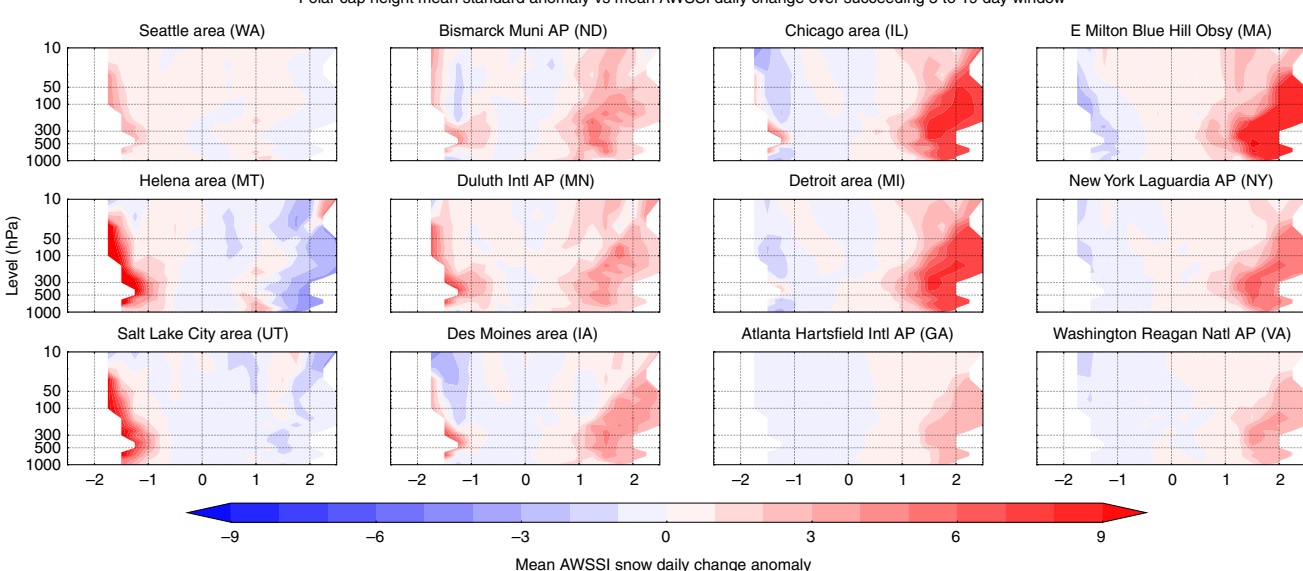

**Fig. 6** Polar cap variability leads increased severe weather up to 19 days. The average daily change in the AWSSI at selected weather stations across the US associated with polar cap geopotential height anomalies (PCH) from the surface to the mid-stratosphere (10 hPa) during 5–19 days preceding the AWSSI values, 1950–2016

Our analysis to this point has been independent of AA, a metric that relates Arctic warming to that of lower latitudes, which may, itself, be caused by sea-ice loss and Arctic-only warming. It is important to note that metrics of PCH and PCT are Arctic-only metrics. While our analysis thus far suggests that a warmer Arctic relative to an Arctic mean temperature is related to colder temperatures across the continents of the NH mid-latitudes, additional analysis is performed to elucidate the degree to which Arctic warming relative to lower latitudes is associated with AWSSI.

The daily increment to the seasonal value of the AWSSI is composited for all standardized PCH for the sub-periods before (1950–1989) and during (1990–2016) the accelerated period of AA (Supplementary Figure 2). The analysis was repeated for the standardized PCT for both before and after the period of AA (Supplementary Figure 3). The relationship is fairly consistent throughout the troposphere. During both periods, the relationship is qualitatively the same: severe winter weather is more common when PCHs/PCTs are elevated throughout the Arctic troposphere. Dividing the winters into early and late winter produced a stronger relationship between PCT and severe winter weather during late winter (Supplementary Figure 4). In our analysis, however, the relationship between a warm Arctic and severe winter weather is stronger in the era prior to AA. Therefore, the analysis presented in Supplementary Figures 2 and 3 indicates that a warm Arctic is related to increased severe winter weather but only suggestive that AA is contributing to increased severe winter weather. One possible exception is in the stratosphere where the relationship between a warm polar stratosphere and increased severe winter weather has become more robust in the period of AA. These findings suggest a growing dependence of severe winter weather in mid-latitudes on weakening of the SPV or SSW (which are defined at 10 hPa) events during the AA era.

**Snowfall before and during era of Arctic amplification.**. Northeastern US cities have experienced a streak of winters with heavy snowfalls over the past two decades, with some famously nicknamed as Snowpocalypse[44], Snowmaggedon[45], and Snow-zilla[46]. Modeling studies have reported divergent conclusions as to whether AA contributes to less[42] or more snowfall[30]. We

computed the return period of varying thresholds of snowfall across the US before (1950–1989) and after (1990–2016) the emergence of AA (Fig. 9). Consistent with our earlier results that a warmer Arctic favors heavier snowfalls, we find that across the northeastern US, heavy snowfalls are generally more frequent since 1990, and in many cities the most extreme snowfalls have occurred primarily during recent decades. In contrast, severe snowfalls in the western US have in general decreased during the AA period. For most cities shown in Fig. 9, the snowfall return periods were found to differ between the two periods with a confidence level greater than 95%.

These findings suggest that recent observed heavy snowfalls, in particular in the northeastern US, may be linked to AA, though further research is required to confirm the linkage. This result is consistent with evidence that extreme rainfall in the northeastern US has also increased over the same period[47].

**Study limitations**. There are important limitations to this study. The most obvious is common to all observational analysis, i.e., correlation does not mean causation. Thus, even though elevated heights and warmer temperatures in the Arctic are positively correlated with more frequent severe winter weather in the mid-latitudes, we cannot conclude that the warmer Arctic is responsible. That said, the highest correlations occur when Arctic variability leads AWSSI by five days, implying it is more likely that Arctic variability is contributing to mid-latitude winter extremes. Another challenge is that the observational record during rapid Arctic change is short, which makes the demonstration of statistical significance difficult[6,8]. We have partially compensated for the short record by computing daily rather than seasonal correlations. This has allowed us to greatly expand the degrees of freedom in analyses of relationships between PCH/PCT and severe winter weather, resulting in highly significant correlations. Based on observations and correlations alone, we also cannot offer physical mechanisms for the relationships we demonstrate, though our analysis is consistent with previously studied mechanisms on how a warmer Arctic can influence mid-latitude weather[5,6,15]. We hope to continue this work with model simulations to identify mechanisms behind the correlations.

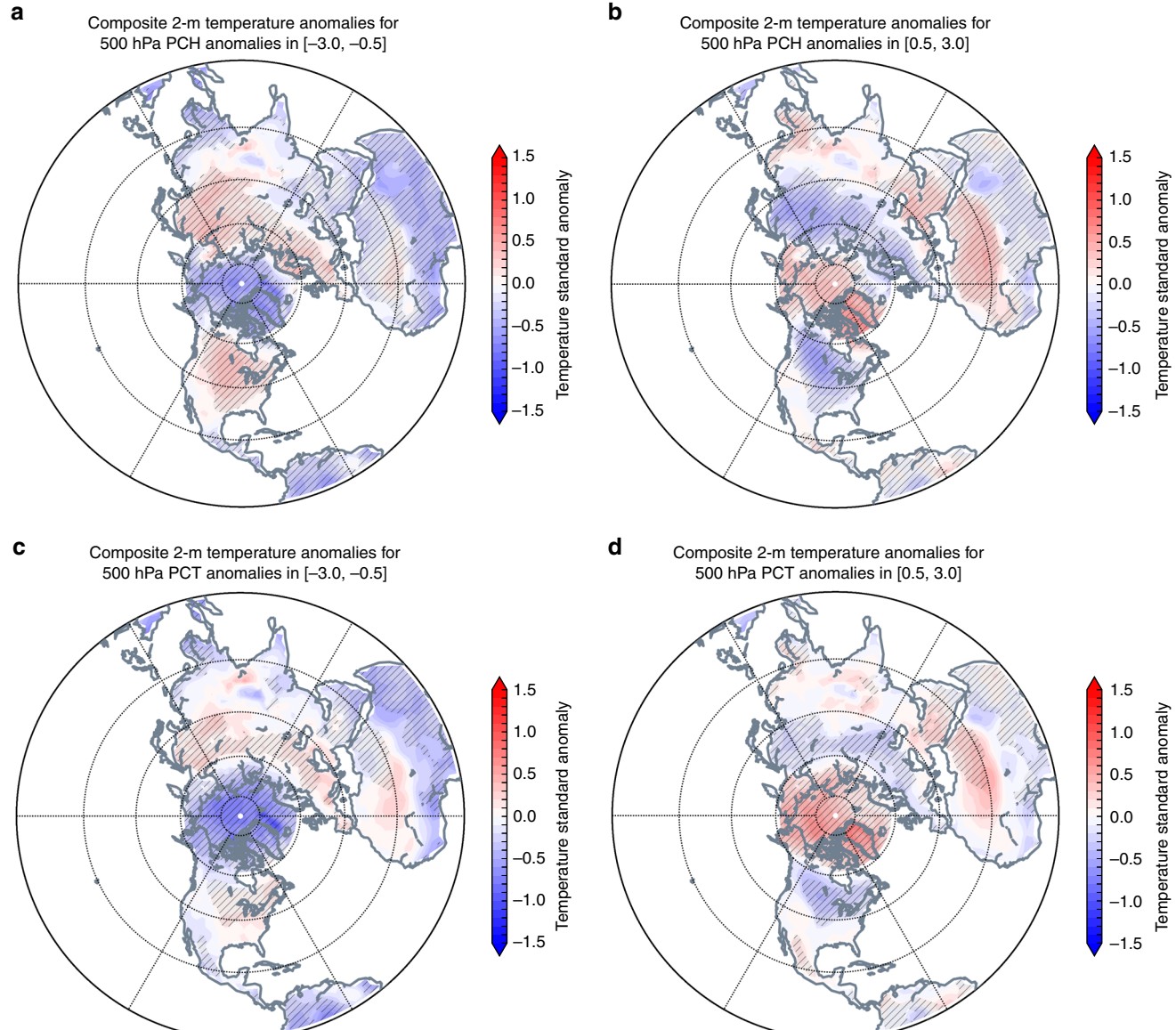

**Fig. 7** As the Arctic warms the continents become colder. Northern Hemisphere surface temperature anomalies plotted for 500 hPa PCH anomalies binned on the intervals **a** [−3.0, −0.5], **b** [0.5, 3.0] and 500 hPa PCT **c** [−3.0, −0.5], and **d** [0.5, 3.0] for all winters 1950–2016. Climatological averages computed over the period 1981–2010. Where difference was found to be statistically significant above 95% is hatched in light gray (e.g., [−3.0, −0.5] to [0.5, 3.0]). We also tested for field significance in all plots and the differences were found to be highly significant. Ocean mask was applied south of 60° N

**Comparing polar cap with the annular mode**. The PCH has been shown to be highly correlated with the first empirical orthogonal function of geopotential height (sea-level pressure) poleward of 20° N, referred to as the Northern Annular Mode: NAM[31,48] (Arctic Oscillation: AO). We now explore the similarities and differences in relationships between the PCH/PCT, the NAM, and severe winter weather.

When the PCH was first introduced[49], it was argued that the advantage of using the PCH over the NAM is that the PCH (and PCT) represents Arctic-only variability, while the NAM includes variability from the entire NH. This distinction is important because the NAM represents a conflation of variability in the Arctic, mid-latitudes, and sub-tropics, thus by definition will exhibit a relationship with mid-latitude weather. Because PCH and PCT are largely independent of mid-latitude influences, any connection between them will shed light on Arctic/mid-latitude linkages.

Though the PCH and the NAM are correlated[48] (Supplementary Figure 5), there are also important differences. By definition, a positive PCH indicates above-normal geopotential heights throughout the Arctic (Supplementary Figure 6). The negative phase of the NAM, in contrast, is characterized by above-normal geopotential heights mostly in the North Atlantic sector, especially near Greenland[50].

The PCT is more strongly related to the PCH, particularly in the troposphere, than it is to the NAM (Supplementary Figure 5). The reason for the independence between the PCT and the NAM in the troposphere can be understood by analyzing surface temperatures ($T_s$) associated with the negative NAM and the positive PCT at 1000 hPa (Fig. 10). We computed the $T_s$ variability related to the NAM at 1000 hPa (Fig. 10a). We also computed the surface temperature variability related to the NAM at 500, 300, and 10 hPa, and the results were qualitatively similar. We further assessed the difference in surface temperature anomalies between the positive

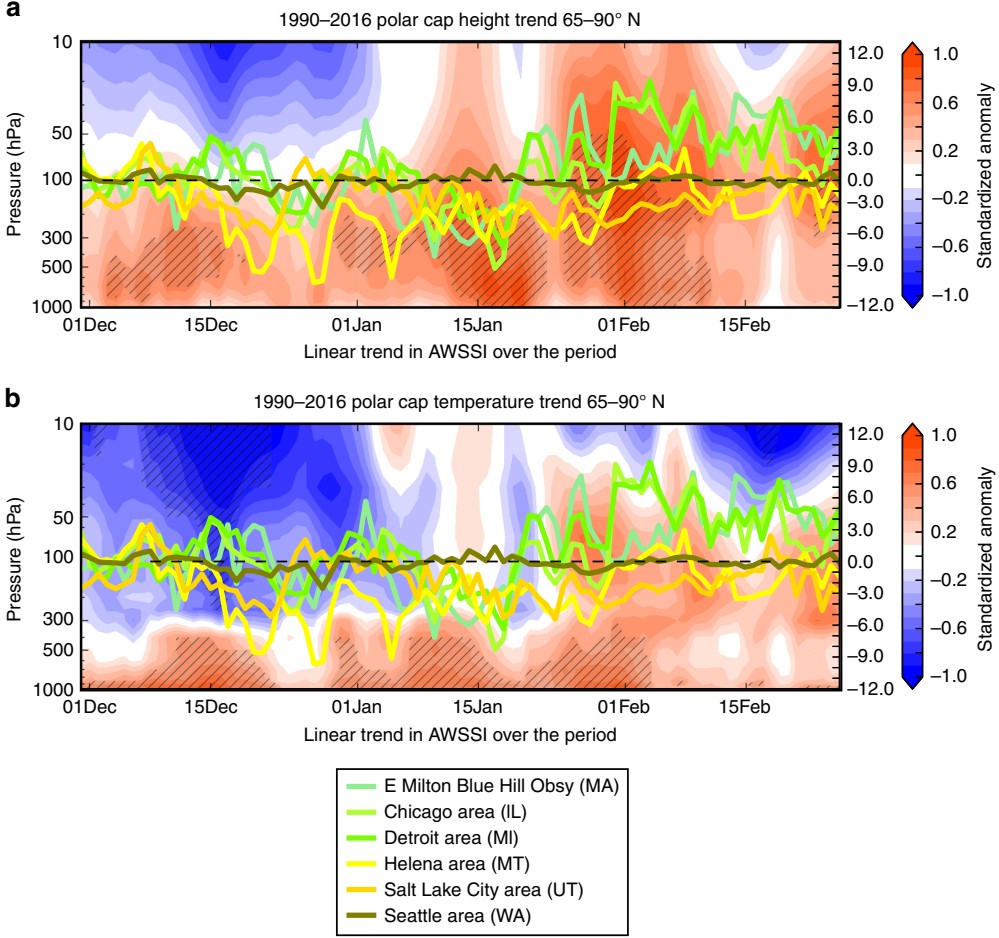

**Fig. 8** Warming trend in Arctic coincides with increased severe winter weather. The annual daily trend in the **a** PCH (shading) and **b** PCT (shading) from the surface to the mid-stratosphere (10 hPa) and the annual trend in the daily change in the AWSSI for three eastern US cities (near Boston, Chicago, Detroit) and three western US cities (Helena, Salt Lake City, Seattle) for the winters 1990/91–2015/16 multiplied by the total number of winters. Statistical significance above 90% for PCH and PCT trends are hatched in dark gray. In Supplementary Figure 8 we have included the variability in the AWSSI with the daily trend

PCT (PCT+) and the negative phase of the NAM (NAM−; Fig. 10b). The temperature anomalies associated with NAM− are positive only near and to the south of Greenland, along with negative anomalies across the Eurasian sector of the Arctic (Fig. 10a). In contrast, the $T_s$ anomalies associated with PCT+ are pan-Arctic and $T_s$ are warmer in the Eurasian Arctic relative to the NAM− (Fig. 10b). The PCT+ represents basin-wide warming, while the NAM− represents mainly regional warming in the Arctic. Also, $T_s$ are colder in the eastern US for PCT+ relative to NAM−. Consistent with this finding we repeated the analysis in Fig. 2 with the NAM index instead of the PCH/PCT in Supplementary Figure 7. Our analysis shows that in general the PCH/PCT have a more robust signal with extreme weather than the NAM. Another difference is that the NAM signal is stronger in the stratosphere in contrast to the PCH/PCT where the tropospheric signal is stronger.

Observed trends in NH surface temperature during the era of amplified Arctic warming are characterized by general warming across the entire Arctic basin punctuated by two regional maxima near Greenland and the Barents–Kara seas (Fig. 10c). The distinct patterns in the NAM versus the PCT suggest that the basin-wide positive temperature trends observed in the period of AA are more similar to, and thus more closely related to, PCT+ than with the NAM−. This is consistent with findings that the influences of the NAM and AA on the NH circulation are distinct in idealized modeling studies[51].

In Fig. 10d we present the 2-m temperature trends associated with temperature anomalies at 850 hPa averaged over the Barents/Kara seas. A broad area of warming extends from the Barents/Kara area eastward through the Eurasian Arctic, while cooling appears elsewhere in the Arctic. Also in contrast to the PCT+ and the NAM− patterns, continental cooling is limited to Asia, along with warming across North America. Temperature trends appear to result from a combination of basin-wide warming, as seen in the positive PCT patterns, and warming in the Barents–Kara region. Thus, constructive interference of these two warming areas may explain the observation of statistically significant continental cooling in Asia only[52]. Basin-wide Arctic warming favors a cooling response in the eastern US, while Barents/Kara warming favors the opposite[35], resulting in destructive interference and partial cancellation (though the temperature trends are still negative). This hypothesis requires further analysis.

In conclusion, from Fig. 10, the PCH and the PCT represent coherent Arctic temperature variability that is basin-wide, while the NAM also varies coherently with these indices but represents regional Arctic temperature variability, which is strongest in and around Greenland. Therefore, the NAM− captures some of the regional warming observed over the past two to three decades (near Greenland), while the PCH+ and PCT+ better capture pan-Arctic warming.

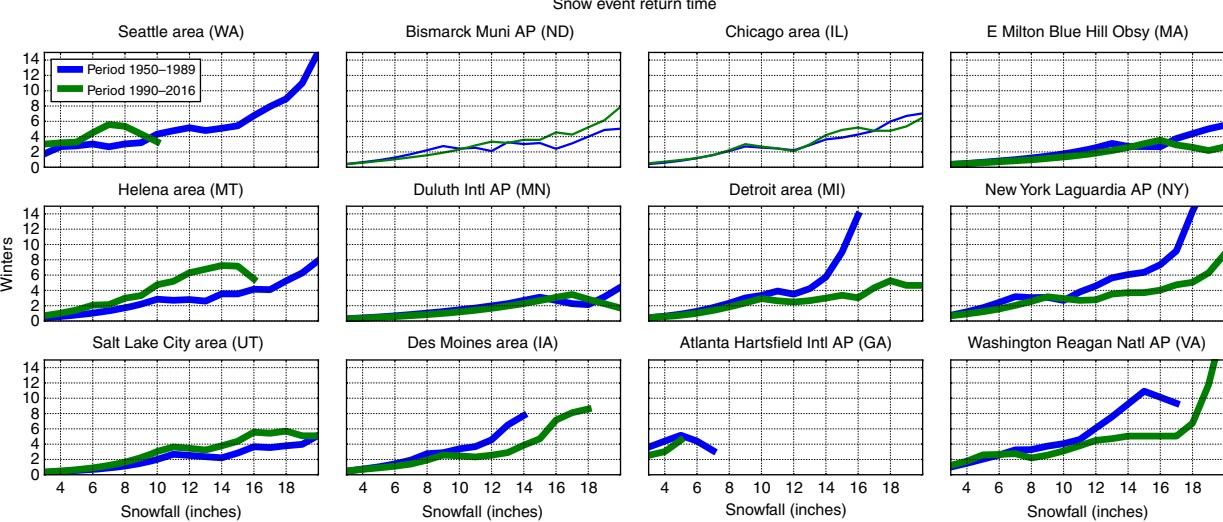

**Fig. 9** Major snowfalls in eastern US are becoming more frequent. The return period (y axis; 0 to 15 years) of varying snowfall events (x axis; 0 to 18 inches) for weather stations during two periods: cold Arctic (1950–1989; blue) and warm Arctic (1990–2016; green). Lower values indicate more frequent snowfalls (shorter return period). The time series that were found to be significantly different at the 95% confidence level are shown in bold lines and include Atlanta, Boston (Blue Hill), Des Moines, Detroit, Helena, New York, Salt Lake City, Seattle, and Washington

## Discussion

We have analyzed two metrics of Arctic variability to demonstrate that an Arctic that is relatively warm is associated with an increase in severe winter weather across the continents of the NH, and in particular, the eastern US. PCH+ and PCT+ in the troposphere, along with PCH+ in the lower stratosphere, are correlated with high values of the AWSSI, a severe weather index that includes cold spells and heavy snowfalls. Based on our analysis, we found that in the lower stratosphere to mid-troposphere (70 to 500 hPa), that PCH+ of two standard deviations or greater is associated with a twofold to fourfold increase in the likelihood of winter weather extremes. These extremes were typically on the order of two to six standard deviations based on the AWSSI. This relationship is most apparent in the northeastern and upper mid-western US.

Although we have not offered mechanistic explanations for these relationships, our findings are consistent with previous studies linking a warming Arctic with extreme winter weather in NH mid-latitudes. Most theories begin with melting sea ice[14,32,53,54], as the past 10 years have exhibited the lowest minimum sea-ice extents since satellite observations began, and sea-ice extent is a relatively easy variable to manipulate in models. The observed increase in autumn snowcover on high-latitude continents is another possible contributor to extreme winter weather, especially across Eurasia[6]. Both less extensive Arctic sea ice and more extensive fall snowcover are related to a warmer Arctic and colder East Asia[18].

Most of the proposed mechanisms linking reduced sea ice and/or increased Eurasian snowcover to extreme winter weather across mid-latitude NH continents involve a pathway through the SPV[6,24,25,32,33,55–57]. Boundary-forcing owing to sea-ice loss and more expansive snowcover can interact constructively with climatological large-scale waves to enhance wave activity and increase energy transfer from the troposphere to the stratosphere, which can trigger a SSW and weaken the SPV. Polar airmasses then spill southward into mid-latitudes, first in the stratosphere and also later in the troposphere[6,15]. Figures 2 and 6 are consistent with a stratospheric pathway and weakened polar vortex as one possible dynamical pathway between Arctic variability and mid-latitude weather. Other proposed mechanisms confine the

Arctic's influence on large-scale circulation changes to the troposphere, in which a warmer Arctic favors a wavier flow and more persistent atmospheric blocking, which often spawns extreme weather events[58,59].

How does this new analysis inform the debate as to whether AA in general and sea-ice loss in particular are contributing to more extreme winter weather? Our focus on two Arctic-only indicators and an index for extreme winter weather offers new evidence and clues about the effects of a rapidly warming and melting Arctic on the rest of the globe. We find that a warmer Arctic atmosphere contributes to dilated geopotential heights locally accompanied by lower heights across mid-latitudes and an equatorward-shifted jet stream. This allows Arctic airmasses to expand farther south while increasing the likelihood of heavy snowfalls. We find a distinction between early winter, when Arctic warming tends to affect only the lower troposphere, and mid-winter to late-winter when PCH+ is evident throughout the troposphere and lower stratosphere. When the entire Arctic atmospheric column is affected, the probability of severe winter weather in mid-latitudes increases, as observed during the era of AA in late winter. Colder Arctic conditions elicit the opposite response. These findings suggest that the continuation of rapid Arctic warming and melting contribute to more frequent episodes of severe winter across the NH mid-latitude continents.

While much has been learned in recent years about direct connections between climate change and weather patterns, uncertainties remain, especially in terms of indirect linkages. Recent work has revealed a variety of possible mechanisms, yet some new studies conclude that a warming Arctic does not force robust cooling over mid-latitude continents, and that recent trends can be explained solely by internal variability[60–63]. Furthermore, climate model simulations that realistically simulate AA indicate that cold extremes and heavy snowfall will decrease as the Arctic continues to warm[41,64,65]. The discrepancies between observational and modeling studies, and also among modeling studies, are recognized[6,8,66,]67 but not well-understood[9,68]. Owing to the important and costly ramifications of changing weather patterns—particularly extreme weather—on society, research should continue rapidly to elucidate the sources of uncertainty in these linkages.

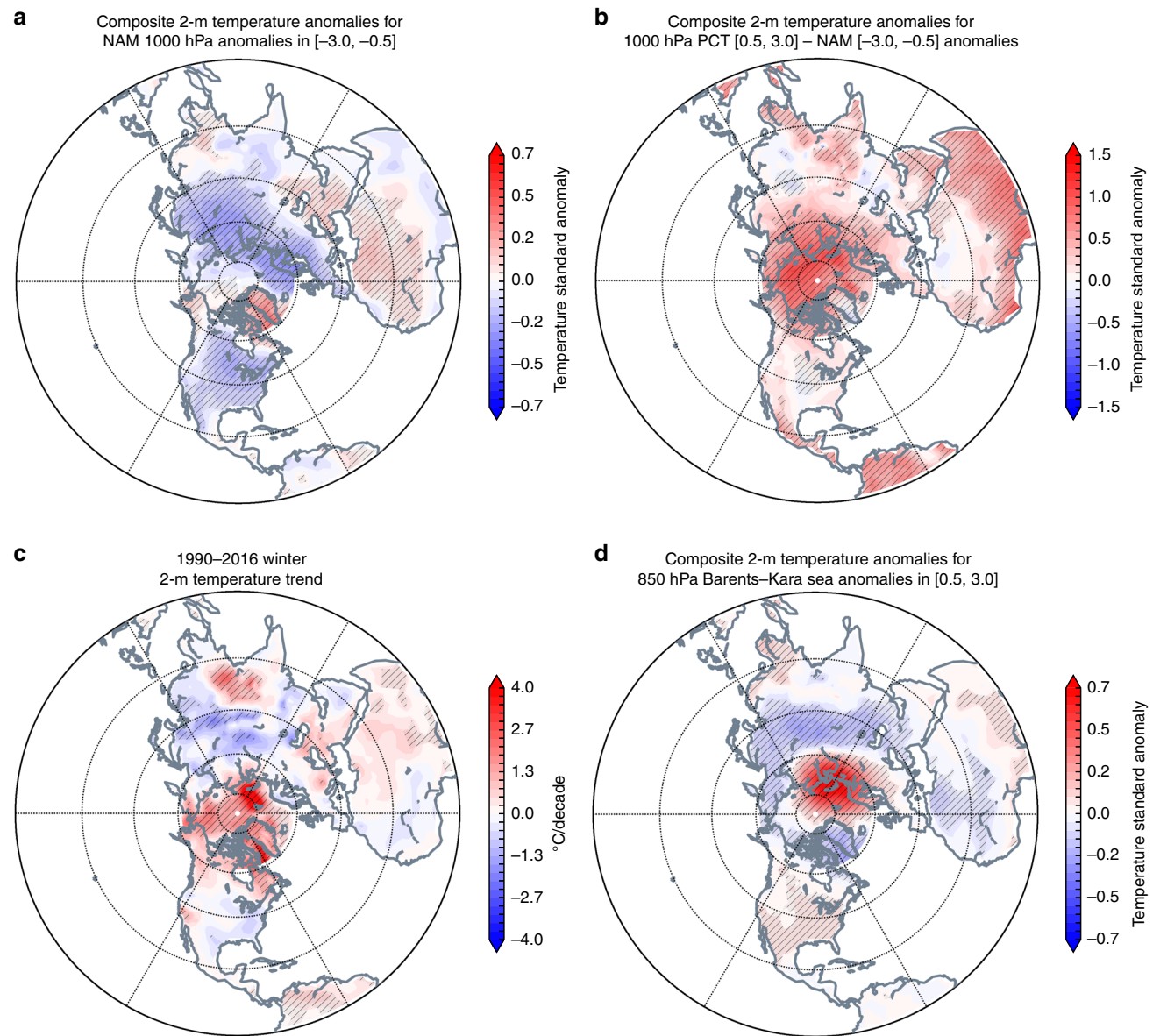

**Fig. 10** Arctic amplification is more closely associated with polar cap temperature than annular mode or warming in the Barents–Kara seas. **a** Surface temperature anomalies associated with the negative phase of the NAM, **b** difference in surface temperature anomalies associated with positive PCT at 1000 hPa and the negative NAM, **c** Northern Hemisphere surface temperatures trends in era of AA (1990–2016), **d** association between surface temperature anomalies across the NH and in the Barents–Kara seas. Climatological averages computed over the period 1981–2010. Note differences in scales. Hatching in all figures represents those values found to be statistically significant above 95%. We also tested for field significance in plots **a**, **b**, and **d** and the differences were found to be highly significant. Ocean mask was applied south of 60° N

## Methods

**Computation of polar cap and severe weather indices**. For this study, the area-weighted polar cap geopotential height and air temperature anomalies (PCH and PCT) were computed for the months December through February using the National Centers for Environmental Prediction (NCEP) reanalysis at global 2.5° resolution for latitudes north of 65° N, using the period 1950 to 2016 for analysis[69]. Daily anomalies were computed from the 1950 to 2016 daily values using the long-term daily mean and standard deviation from the reference period 1981 to 2010.

We analyzed the AWSSI for multiple stations across the United States. The AWSSI is calculated using several variables[29] and is available from http://mrcc.isws.illinois.edu/research/awssi/indexAwssi.jsp. Of those stations analyzed, we chose 12 geographically representative stations across the United States. Those stations are displayed on a topographic map created with the open source package matplotlib (https://matplotlib.org) in Fig. 1. In this study, we analyzed the AWSSI (Fig. 2) and the individual components of the AWSSI of surface temperature (Fig. 3) and snowfall (Fig. 4). The Student's *t* test was applied to the bins pairwise as described in the text, adjusting downward the degrees of freedom based on the autocorrelation in the time series.

PCH and PCT were compared to departures from the mean in AWSSI values at individual stations over winter months (DJF) from 1950 to 2016. AWSSI changes were binned (0.25 standard deviation bin width) based on corresponding polar cap anomalies evaluated from 1000 to 10 hPa. To compute statistical significance, data used in Fig. 2 and later in Figs. 7 and 10 were detrended by removing the linear trend in the time series prior to binning the data. Statistical significance was evaluated with a two-tailed Student's *t* test comparing the positive anomaly bins to the negative anomaly bins and degrees of freedom were adjusted downwards based on the autocorrelation in the time series. Most relationships between the polar cap variables and winter weather were found to be almost uniformly statistically significant at $p < 0.05$. Bin counts were generally well-balanced and on the order of 2000 to 3000 days total across all bins in each of the positive and negative spaces, for total counts typically 4000 to 6000 days per station (Fig. 2).

**Lagged correlation analysis**. Lagged time analyses were used to demonstrate the direction of the relationship between positive PCH values and more frequent severe winter weather. Single-day correlations were computed at lags from −30 to +30 days for AWSSI change across all isobaric levels (500 hPa depicted in Fig. 5).

To further explore the direction of this relationship, we analyzed the average daily increment to the seasonal value of the AWSSI at all stations following PCH anomalies from 1000 to 10 hPa, computed for preceding lags by 1 to 4 weeks, with the 5 to 19 day analysis depicted in Fig. 6.

**Computation of hemispheric variables**. We expanded the analysis from the US-based AWSSI to the entire NH by analyzing NCEP reanalysis surface temperature anomalies with respect to PCH and PCT values (Fig. 7) at all levels. For hemispheric plots (Figs. 7 and 10), PCH and PCT changes were binned at 0.5 standard deviation bin width. The 500-hPa level was chosen for depiction because of the relatively strong response in AWSSI at these levels (Fig. 2). Paired-sample tests of the spatial time series were used to identify regions of statistical significance at $p < 0.05$. We also computed local field significance to determine that differences are significantly different[70].

**Trend analysis**. The linear daily trend in both PCH and PCT was computed during DJF for the period of record, and this trend was then compared with trends in the daily AWSSI changes throughout the same period; this analysis is depicted in Fig. 8 for selected stations. Stations were selected to provide good geographic representation of the US with large population centers given preference.

**Computation of snowfall returns**. To extend the analysis of snowfall related to polar cap anomalies (Fig. 4) we examined specific snowfall trends over the periods 1950 to 1989 and 1990 to 2016. Snowfall events were defined as occurring across consecutive days with non-zero and non-trace snowfall. For example, 3 days of 4-inch snowfall would be analyzed as a 12-inch snow event. Return times for snow events of 1 to 18 inches were computed across the period in days, and then transformed to winters based on a 90-day winter but included all 365 days of the year when computing return times. So, for example, a 3-inch snowfall may occur four times in a single winter, about 90 days with a return time of 22.5 days. An 18-inch snowfall may occur roughly once every two winters, or 730 days. We talked through several strategies to describe this effectively and the return time as plotted was the outcome. We used a Wilcoxon non-parametric test for statistical significance in snowfall return, given the discontinuous nature of snowfall events and no reason to believe variances would be equal between 1950–1989 and 1990–2016 partitions. Using this metric, the snowfall returns were statistically significant at $p < 0.05$ for Seattle, Helena, Salt Lake City, Des Moines, Atlanta, Duluth, Washington, New York City and Boston (Blue Hill). These results are depicted in Fig. 9.

**Computation of hemispheric analysis and trends**. We extended the linear trend analysis (Fig. 8) with a hemispheric analysis of 2-m temperatures using the period of AA (1990–2016) in the NCEP Reanalysis data set (Fig. 10c). These 2-m temperature anomalies were then examined as a composited analysis with the negative NAM at 1000 hPa, binning temperature anomalies with NAM anomalies in the interval [−3.0, −0.5] (Fig. 10a). NAM changes were binned at 0.5 standard deviation bin width. The separation of the negative NAM from the positive polar cap anomalies are emphasized with a difference analysis comparing these binned temperature anomalies between the 1000 hPa PCT composite and the negative NAM composite (Fig. 10b). The 1000 hPa level was used for the PCT analysis to compare with the NAM at the same level and as a close approximation of the AO which is computed with sea-level pressure. These same 2-m temperature anomalies were evaluated with respect to the Barents–Kara Sea region (65 to 80° N, 10 to 100° E) at 850 hPa (Fig. 10d). We also computed local field significance for Fig. 10a, b to determine that differences are significantly different[70].

**Inclusion of El Niño/Southern Oscillation**. To extend the time series analysis in Fig. 5, a similar analysis was performed using monthly values of the Niño 3.4 index and composited AWSSI values for DJF to assess the strength of that relationship (Supplementary Figure 1). For this analysis, the Niño 3.4 index was obtained from the NOAA Earth Systems Research Laboratory Physical Sciences Division.

**Inclusion of Northern Annular Mode**. The NAM data set used in these analyses was obtained from the website of Northwest Research Associates (NWRA) and computed by Dr. Mark Baldwin (http://www.nwra.com/resumes/baldwin/nam_index_1958-2006.zip). A preliminary correlation analysis was computed for both PCH and PCT from 1000 to 10 hPa for the period of record for these data (Supplementary Figure 5a, b). For this same period of record (1958–2006), the correlation of PCT with PCH was computed (Supplementary Figure 5c) for comparison. To further explore the strength of the relationship between polar cap anomalies and NAM anomalies, composite analysis was made with the Northern Hemisphere 500 hPa geopotential height anomalies and 500 hPa PCH anomalies (Supplementary Figure 6a) and 500 hPa NAM anomalies. The difference between composited PCH and NAM anomalies (Supplementary Figure 6b) further highlights the distinction between polar cap and NAM variability. We also repeated the analysis shown in Fig. 2 but substituted the NAM for PCH/PCT to compare the relationship between the NAM and PCH/PCT with winter weather variability in Supplementary Figure 7.

**Computed variability of severe weather index**. Finally, as a measure of the uncertainty of the daily AWSSI trends shown in Fig. 8, we computed the daily standard deviation of the AWSSI for the three eastern and three western US cities shown in Fig. 8 and include the ±1 standard deviation with the daily trend for 1990 to 2016 of the AWSSI in Supplementary Figure 8.

**Data availability**. All data used in our analysis is publicly available and sources provided. All derived indices used in the analysis is available upon request from corresponding author.

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

## Acknowledgements

We are grateful to Barbara Mayes-Boustead and Steve Hallberg for generously sharing with us the AWSSI data. J.C. is supported by the National Science Foundation grants AGS-1303647 and PLR-1504361. J.F. is supported by NASA grant NNX14AH896 and NSF/ARCSS grant 1304097.

## Author contributions

J.C. and K.P. performed the analysis. J.C. and J.F. wrote the manuscript.

## Additional information

**Competing interests:** The authors declare no competing financial interests.

