## [Peer Review File · Nature Communications]

Reviewers' comments:

Reviewer #1 (Remarks to the Author):

Review comments for "Warm Arctic episodes linked with extreme winter weather in Northern Hemisphere mid-latitudes" by Cohen et al.

Recommendation: Major revision

This paper offers observational evidences of climate linkage between Arctic and mid-latitude winter weather through introduction of Accumulated Winter Season Severity Index (AWSSI), which is recently developed for a measure of wintertime severity over U.S. Some perspectives from this paper provide new knowledge about Arctic climate impacts on winter weather property over the U.S. Analyses were well conducted to provide results evidential for authors' arguments. Discussions are well organized. However, overall the novelty and scientific impacts from this paper seem to be not so high, because the statistical relationship of winter Arctic and mid-latitudes has already reported by many studies. Notable argument is that Arctic-only metrics can capture recent Arctic amplification (AA) property better than NAM index. This is interesting and novelty of this paper. I think this paper has a potential to be published in Nature Communications if the advantage of this paper were more extended. I hope my comments and questions listed below will help to improve the manuscript.

Major comments:

1) This paper employed AWSSI for a measure of wintertime severe weather occurrence. This index is obtained using max/min temperature and snow fall/depth observed at widely distributed weather stations over U.S. While I briefly read a paper of Mayes-Boustead et al. (2015), I feel it is somewhat difficult to give a scientific significance to this index, because point thresholds are arbitrary and weight ratio between temperature and snow vary among stations and years (from their section 3. b and F5 and F8).

I think that at least authors should mention about a motivation of use of this index instead of usual weather variables.

Mayes-Boustead, B. E., S. D. Hilberg, M. D. Shulski, and K. G. Hubbard (2015), The Accumulated Winter Season Severity Index (AWSSI), *Journal of Applied Meteorology and Climatology*, 54, 8, 1693-1712.

2) Authors claim that PCH/PCT well captures AA property while NAM does regional, through comparison between PCH/PCT and NAM at 1000 hPa level (L239-307). Further, authors claim that southward expansion of Arctic cold airmass is associated with recent AA property (i.e., high Arctic and low mid-latitudes) (L344-347). I think such the circulation change associated with meridional pressure seesaw is a major property of AO/NAO/NAM (e.g., Yu et al., 2015; Nakamura et al., 2016). Why is PCH/PCT better than NAM1000? If NAM500, 300, and 10 were used instead of NAM1000, will comparison results of PCH/PCT vs NAM change?

Yu, Y., R. Ren, and M. Cai (2015), Comparison of the mass circulation and AO indices as indicators of cold air outbreaks in northern winter, *Geophys. Res. Lett.* 42, 2442–2448, doi:10.1002/2015GL063676.

Nakamura, T., K. Yamazaki, M. Honda, J. Ukita, R. Jaiser, D. Handorf, and K. Dethloff (2016), On the atmospheric response experiment to a Blue Arctic Ocean, *Geophys. Res. Lett.*, 43, 10,394 – 10,402, doi:10.1002/2016GL070526.

3) Furthermore, authors argue importance of the stratosphere consistent with previous studies (L326-336, L347-351). Daily NAM index is widely used tool to diagnose the intensity of the stratospheric polar vortex. It is provided at multiple levels between 1000 and 10 hPa, so that authors can apply same analysis of this paper (e.g., F1-F5) using NAM index instead of PCH/PCT. If do so, will the results more clarify the stratospheric role?

In association with comments 2) and 3);

Argument of that Arctic-only metrics PCH/PCT are better capturing recent AA property than NAM index is very interesting and novel, because I have thought NAM index already captures warming Arctic and cooling continents signals. To confirm this argument more robustly, I strongly recommend that authors repeat the analyses using multi level NAM index and compare with PCH/PCT results.

Even if NAM at middle tropospheric level well captured the recent AA property than PCH/PCT, I believe that strong relationship between Arctic-only metrics and mid-latitudes severe weather shown in this study remains valuable.

Minor comments:

Methods

L367: Reanalysis data errors are expected large in Polar region. NCEP data is relative old generation reanalysis. How much is errors among reanalysis in the Polar region? Is it sufficiently as small as the overall results of this study will be unchanged if the other reanalysis data (e.g., JRA55) were used?

L373 about AWSSI: see my major comment 1).

L379 "AWSSI changes were binned ..." : How much is bin width?

L412-414 "For example, a return time of two winters would be on the order of 700 days, while 4 events per winter corresponds to a return time of 20 days." : I was confused by this sentence. Is there a gap between the estimated return times exceeding a winter season (1 time per 2 winters is about 700d) and shorter than a season (4 times per 1 winter is not 90d but 20d)?

Results

L69: What is "daily change in AWSSI"? Day-to-day anomaly, anomaly of daily score from its seasonal mean, or anomaly from daily climatology?

L87-88 "temperatures are therefore contributing more strongly to the relationship between PCH and AWSSI." : I am not convinced because AWSSI scores arbitrary function that weights extreme value of temperature and snowfall and originally the contributions of temperature and snowfall scores are not even.

101: in Arctic temperature in the troposphere.

L120: PCH at what level?

L137-144: Interpretation may depend on definition of "daily change in AWSSI". If it is a day-to-day anomaly or anomaly from seasonal mean, its time scale is too short against ENSO's time scale. Furthermore, for a comparison of Arctic vs tropical influence, is it better to use PNA index instead of Nino3.4 anomaly, because PCH/PCT are not boundary forcing itself?

L148: for PCH, is unit a meter?

L183: In contrast, in the eastern US. ...

L196-204: There seems to be notable difference between pre- and during AA era as that relationship is stronger in pre-AA (top panels of Supp. F2 and F3) than in AA era (bottom panels) even in the stratosphere. So I could not understand "a growing dependence of severe winter weather in mid-latitudes on weakening of the stratospheric polar vortex or SSW events during the AA era" (L203-204).

L231-233 "This has allowed us to greatly expand the degrees of freedom in analyses of relationships between PCH/PCT and severe winter weather, resulting in highly significant correlations." : Did authors consider the duration of daily time series and appropriately decrease the degrees of freedom for a calculation?

L277 & Fig9c: Statistical significance of the trend should be given.

L289: In association with this paragraph, Kug et al. (2015) should be cited.

Kug, J.-S., J.-H. Jeong, Y.-S. Jang, B.-M. Kim, C. K. Folland, S.-K. Min, and S.-W. Son, 2015: Two distinct influences of Arctic warming on cold winters over North America and East Asia. *Nat. Geosci.*, 8, 759–762, doi:10.1038/ngeo2517.

L302: As in my major comment 2), if NAM500 were used instead of NAM1000, will results change?

L344-353: I agree in general, but could this explain why the cooling is obvious mainly in the continents? Is it because of difference of radiative and surface heat flux properties in Ocean and land rather than zonal asymmetry potentially involved in the atmospheric mode as AO/NAO?

Reviewer #2 (Remarks to the Author):

Overall the manuscript reads well, and is very clear in what is presented. The statistical relationships presented are easy to understand, and the overall argument is clear. I think these results help further our understanding of AA influence on mid-latitude weather, and spur further work.

The revisions I have suggested are generally minor in scope, and focus on adding more in the way of limitations and also clarifying some of the methods and interpretation.

On the methods:

1. Why were the specific AWSSI stations chosen? In particular, given the substantial spatial variability in snowfall this may affect results.
2. I do grasp that in order to improve sample size, a broader net of what is AA needs to be cast. This said, why is 1990 the start date? It seems like a focus just on the last decade (since 2007) would be insightful, or even a broader period back to the late 1990s.
3. Further, the authors note the trends are most substantial during the typical time of SSW (line 184); did the authors attempt to partition the relationship out by time of the season? That is, to focus on the mid-to-late winter, when the Arctic – mid-latitude relationships seem most robust?

On Limitations:

4. The AWSSI is an interesting and useful index. One issue of concern is that the way it was defined was with a mid-latitude continental climate of the midwest in mind, and hence by its nature it will be most responsive in this region. It is thus in one sense unsurprising to see the strongest relationships in the region surrounding where it was defined. More caution should be used in interpreting the results – in ‘marginal’ places like Atlanta and Seattle, single events are going to have substantial influence on overall values, more so than in the more continental locations. A little more contextualization on what the AWSSI is, and what it conveys and its limitations, would help understand the results better.

Other comments:

5. Line 32-34: There are a lot of studies that do not show widespread cooling on NH continents in recent decades; in Figure 9c in the paper itself, something approaching ‘widespread’ could only be claimed for Asia. It seems to be much more so a lack of increase, and much greater year-to-year variability. This sentence should be modified to reflect this.

6. Paragraph in lines 61-67... it might be best to put the last sentence first, to make it clearer that the three variables are not being independently assessed, but rather it’s the relationship among them that you’re examining.

7. Line 118 – change it’s to it is.

8. Line 199-200: The authors state that the ‘relationship is fairly consistent throughout the troposphere, for both sub-periods...’ In Figure S2, the two periods look vastly different, with an overall weaker relationship in the AA period. What am I missing?

9. Line 211 – should be Figure 8, not Figure 3. With regard to this figure and discussion, are the differences statistically significant?

10. Lines 367-371. Authors should mention the subset of months used here within each calendar year. Also, for anomalies, are these anomalies de-seasonalized too? If so, how?

11. Figures: Blue Hill line points to Philadelphia, and there is no Philadelphia line.

12. Figure 5. I don’t agree with the figure caption “by at least two weeks”, since the authors explore 5-14 days here – thus there is no evidence of the “at least” part.

13. Figure 7. I’d suggest making the eastern city lines one range of hues, and the western ones another, and also organize the legend by east vs. west, to facilitate understanding

14. Figure 9c. Was statistical significance testing done, and nothing was significant? Or was it not done?

Scott Sheridan

Reviewer #3 (Remarks to the Author):

The presented study by Cohen et al. gives new insights into a very interesting and intensively debated hot topic that is the observed “counterintuitive” cooling trend in the NH mid-latitude continents in recent winters despite an otherwise warming world. The authors show that the observed cooling trends and particularly the increased frequency of severe winter weather are associated with a warming Arctic. Their results will be a valuable contribution to the scientific discussion and also interesting for the general reader of Nature Communication.

Following, I have listed three main points regarding method, figures, and general understanding that I would like to see addressed and some minor comments are given below.

(i) Method

I am a little bit concerned about the high and frequent significance of the presented results and the underlying statistical analyses. For example, daily data within one winter season will be highly auto-correlated. The applied t-test, however, requires independent variables. Has this been taken into account when estimating significance? This point is crucial for most of the presented results. As I understand, significance in Figure 1 (and others) has been calculated by comparing negative and positive anomaly bins. But how exactly has this been done? Did you compare all AWSSI values of a given pressure level associated with negative PCH anomalies with all respective AWSSI values associated with positive PCH anomalies? For Figure 1 PCH and PCT have been standardized. But what about AWSSI? I would expect that this time series needs to be at least detrended. In the abstract you mention a robust and linear relationship between PCH/PCT and AWSSI. Do you extract this from Figure 1? If so I would like to see more quantitative evidence for this, particularly, because from the figure itself a linear relationship between PCT and AWSSI seems to be absent for many cities.

Figure 6 and figure 9 should be repeated using field significance testing (see Wilks, BAMS, 2016).

(ii) Figures.

In my print out it was basically impossible to read the text and numbers in those figures showing the US map. I think it would be enough to show the map with the station locations once in the beginning and then use the same ordering of stations for all figures.

Please add units to all x and y axes.

(iii) General understanding.

I found it confusing that on p. 4 the authors refer to "warm Arctic" when in fact they talk about positive PCH anomalies. This is particularly confusing, because in the lower panel of Figure 1 they do show Arctic temperatures. This misunderstanding is partly solved in the paragraph from line 95-101 which I think is very important and should be explained more (including refs), particular, because this may not be clear to the general audience of Nature Communications. Having said this, it would improve readability if the authors clarified from the beginning why they look at PCH and PCT (and not NAM for example).

To me it looks like the correlation between PCH and AWSSI is much stronger in the pre-AA era (Supplementary Figure 2). This should be discussed in more detail since the influence of AA is an important part of the study.

For a quantitative analysis – which the authors claim distinguishes their analysis from others – I would actually expect a sentence like "for positive PCH anomalies X standard deviations above the climatological mean we find a Y% increase in severe winter extremes" or similar.

Minor

- l. 51: I am not perfectly familiar with the literature but somehow I would be surprised if there were no other studies which are not constrained by these limitations.
- l. 57: I would add which pressure levels have been used since this is important for PHC
- l. 59: Poleward of 60°N
- l. 78: probably correct, but just from the figures peak AWSSI could in many cases be somewhere between 1.5 and 2.5 PCH anomaly
- l. 111: here you only show the link to U.S. mid-latitudes
- l. 112: weekend polar vortex in the troposphere or stratosphere or both?
- l. 119: what do you mean by amplified flow?
- l. 185 weather has increased in eastern US.
- l. 211: Figure 3 should be Figure 8
- l. 217: This statement is too strong in my opinion, because the presented analysis is simply based on two different time periods
- l. 341: [...] two Arctic-only indicators and a new index [...]

- l. 412: I did not quite understand how exactly you compute the return times. Also I found the example slightly confusing because return times of “winters” are based on 365 days per year and “events” are based on 90 days per year. Here, again, auto-correlation may be important.
- l. 654: There are many examples where $p > 0.01$
- SI l. 1: difference
- SI l. 21: Fig.1 should be Fig. 2

Response to Reviewers for “Warm Arctic episodes linked with extreme winter weather in Northern Hemisphere mid-latitudes” by Cohen et al.

Reviewer #1

We are grateful to the reviewer for these constructive comments. They have helped improve the manuscript significantly.

Major comments:

1) *This paper employed AWSSI for a measure of wintertime severe weather occurrence. This index is obtained using max/min temperature and snow fall/depth observed at widely distributed weather stations over U.S. While I briefly read a paper of Mayes-Bousted et al. (2015), I feel it is somewhat difficult to give a scientific significance to this index, because point thresholds are arbitrary and weight ratio between temperature and snow vary among stations and years (from their section 3. b and F5 and F8). I think that at least authors should mention about a motivation of use of this index instead of usual weather variables.*

We included in the revised manuscript better motivation for using the AWSSI index. For our study, the AWSSI is advantageous because it integrates both intensity and duration of temperature, snowfall and lying snow into one index to measure weather severity or weather extremes. The singular value is representative of multiple weather parameters that all contribute to severity facilitating the comparison of winter weather extremes across seasons and stations.

2) *Authors claim that PCH/PCT well captures AA property while NAM does regional, through comparison between PCH/PCT and NAM at 1000 hPa level (L239-307). Further, authors claim that southward expansion of Arctic cold air mass is associated with recent AA property (i.e., high Arctic and low mid-latitudes) (L344-347). I think such the circulation change associated with meridional pressure seesaw is a major property of AO/NAO/NAM (e.g., Yu et al., 2015; Nakamura et al., 2016). Why is PCH/PCT better than NAM1000? If NAM500, 300, and 10 were used instead of NAM1000, will comparison results of PCH/PCT vs NAM change?*

We performed the additional analysis requested by the reviewer. Substituting NAM at 500, 300 and 10 mb did not qualitatively change the results (see figure below). During negative AO at all levels warm temperatures in the Arctic on the North American side and cold temperatures on the Eurasian side of the Arctic are observed. We mentioned this in the text.

3) Furthermore, authors argue importance of the stratosphere consistent with previous studies (L326-336, L347-351). Daily NAM index is widely used tool to diagnose the intensity of the stratospheric polar vortex. It is provided at multiple levels between 1000 and 10 hPa, so that authors can apply same analysis of this paper (e.g., F1-F5) using NAM index instead of PCH/PCT. If do so, will the results more clarify the stratospheric role?

We repeated the analysis in Figs. 1-5 with the NAM index. From Figure 1 we would say that the NAM signal is stronger in the stratosphere in contrast to the PCH and PCT that are stronger in the troposphere. Also in general the PCH/PCT have a more robust signal with extreme weather than the NAM. We included the first figure in the Supplementary Information but did not include the others. We do feel that this additional figure helps to further differentiate the NAM vs. the PCH/PCT.

Same as Figure 1 from manuscript but using the NAM index instead of PCH/PCT.

Same as Figure 2 from manuscript but using the NAM index instead of PCH.

Same as Figure 3 from manuscript but using the NAM index instead of PCH.

Same as Figure 4 from manuscript but using the NAM index instead of PCH.

Same as Figure 5 from manuscript but using the NAM index instead of PCH.

In association with comments 2) and 3);

Argument of that Arctic-only metrics PCH/PCT are better capturing recent AA property than NAM index is very interesting and novel, because I have thought NAM index already captures warming Arctic and cooling continents signals. To confirm this argument more robustly, I strongly recommend that authors repeat the analyses using multi level NAM index and compare with PCH/PCT results. Even if NAM at middle tropospheric level well captured the recent AA

property than PCH/PCT, I believe that strong relationship between Arctic-only metrics and mid-latitudes severe weather shown in this study remains valuable.

We are pleased that the reviewer recognizes the value of our work. As it turns out the additional analysis requested by the reviewer further strengthened the differences between the NAM and PCH/PCT.

Minor comments:

Methods

L367: Reanalysis data errors are expected large in Polar region. NCEP data is relative old generation reanalysis. How much is errors among reanalysis in the Polar region? Is it sufficiently as small as the overall results of this study will be unchanged if the other reanalysis data (e.g., JRA55) were used?

Geopotential height analyses are similar across reanalysis datasets and we are averaging over a large domain (65-90°N). Our previous analysis with PCH has shown nearly identical results with different reanalysis datasets. We chose the NCEP/NCAR because it had the advantage of extending back to 1950.

L373 about AWSSI: see my major comment 1).

We attempted to address the issue raised by the reviewer.

L379 "AWSSI changes were binned ..." : How much is bin width?

Each bin width is 0.50 standard deviations. Now included in the text.

L412-414 "For example, a return time of two winters would be on the order of 700 days, while 4 events per winter corresponds to a return time of 20 days." : I was confused by this sentence. Is there a gap between the estimated return times exceeding a winter season (1 time per 2 winters is about 700d) and shorter than a season (4 times per 1 winter is not 90d but 20d)?

We edited the text to make this clearer. We only considered snowfall that occurred for the three months of December through February but included all 365 days of the year when computing return times. So, for example, a 3-inch snowfall may occur four times in a single winter, about 90 days. An 18-inch snowfall may occur roughly every two winters, or 730 days. We talked through several strategies to describe this effectively and the return time as plotted was the outcome.

Yes, there is a gap for the cities considered in this study; we did not look at snowfall outside the DJF period, so if a return time is longer than 90 days it is then 'next season' or 365 days.

Results

L69: What is “daily change in AWSSI”? Day-to-day anomaly, anomaly of daily score from its seasonal mean, or anomaly from daily climatology?

The AWSSI is incremented daily and accumulated throughout the season. We computed the daily value of the AWSSI or the daily change in the seasonal value. We edited the text and hopefully it is now clearer.

L87-88 “temperatures are therefore contributing more strongly to the relationship between PCH and AWSSI.” : I am not convinced because AWSSI scores arbitrary function that weights extreme value of temperature and snowfall and originally the contributions of temperature and snowfall scores are not even.

We removed the text.

101: in Arctic temperature in the troposphere.

We included “troposphere”.

L120: PCH at what level?

At 500 hPa and this is now included.

L137-144: Interpretation may depend on definition of “daily change in AWSSI”. If it is a day-to-day anomaly or anomaly from seasonal mean, its time scale is too short against ENSO’s time scale. Furthermore, for a comparison of Arctic vs tropical influence, is it better to use PNA index instead of Nino3.4 anomaly, because PCH/PCT are not boundary forcing itself?

We agree with the reviewer but variability in the PNA is not simply due to tropical forcing. Also, the PNA has the same limitations as the AO; it is computed using information from the mid-latitudes and therefore should not be used to inform on remote forcing of mid-latitude weather. In addition, given that forecasts are issued for the degree of severity of winter weather dependent on the phase of ENSO therefore we do think that there is justification to analyze the relationship between change in the AWSSI and ENSO variability. We repeated the analysis of relationship of the AWSSI with daily PCH variability but with monthly PCH anomalies so as to match the same time scale of variability as the ENSO index. Our analysis showed that the range of AWSSI variability associated with the monthly PCH is much greater than AWSSI variability associated with the monthly ENSO index. The analysis using ENSO is tangential to the main analysis. We thought that it would be of interest to include it but if the reviewer insists, the figure and accompanying text can be removed.

L148: for PCH, is unit a meter?

The unit is meters but all values of PCH are normalized by their standard deviation. We now include this in the text.

L183: In contrast, in the eastern US. ...

Changed, thank you.

L196-204: There seems to be notable difference between pre- and during AA era as that relationship is stronger in pre-AA (top panels of Supp. F2 and F3) than in AA era (bottom panels) even in the stratosphere. So I could not understand “a growing dependence of severe winter weather in mid-latitudes on weakening of the stratospheric polar vortex or SSW events during the AA era” (L203-204).

We agree with the reviewer that in the troposphere and even the lower stratosphere the relationship is stronger pre-AA than during the era of AA. However, that is not true in the mid-stratosphere (10-50 hPa) where the relationship between severe winter weather and a weak vortex is stronger during the era of AA than pre-AA. We now indicated in the text that we are referring to the mid-stratosphere.

L231-233 “This has allowed us to greatly expand the degrees of freedom in analyses of relationships between PCH/PCT and severe winter weather, resulting in highly significant correlations.” : Did authors consider the duration of daily time series and appropriately decrease the degrees of freedom for a calculation?

In the revised manuscript, we accounted for auto-correlation when computing the statistical significance.

L277 & Fig9c: Statistical significance of the trend should be given.

We now included the statistical significance of the trend in Figure 9c.

L289: In association with this paragraph, Kug et al. (2015) should be cited. Kug, J.-S., J.-H. Jeong, Y.-S. Jang, B.-M. Kim, C. K. Folland, S.-K. Min, and S.-W. Son, 2015: Two distinct influences of Arctic warming on cold winters over North America and East Asia. Nat. Geosci., 8, 759–762, doi:10.1038/ngeo2517.

Citation now included.

L302: As in my major comment 2), if NAM500 were used instead of NAM1000, will results change?

The results did not change but see our full comments above.

L344-353: I agree in general, but could this explain why the cooling is obvious mainly in the

continents? Is it because of difference of radiative and surface heat flux properties in Ocean and land rather than zonal asymmetry potentially involved in the atmospheric mode as AO/NAO?

This is an interesting point raised by the reviewer but we feel that it is beyond the scope of the manuscript to address properly.

Reviewer #2 (Remarks to the Author):

We are very grateful to Dr. Sheridan for these constructive comments and suggestions that have improved the manuscript significantly.

Overall the manuscript reads well, and is very clear in what is presented. The statistical relationships presented are easy to understand, and the overall argument is clear. I think these results help further our understanding of AA influence on mid-latitude weather, and spur further work.

The revisions I have suggested are generally minor in scope, and focus on adding more in the way of limitations and also clarifying some of the methods and interpretation.

We thank the reviewer for his encouraging words.

On the methods:

1. Why were the specific AWSSI stations chosen? In particular, given the substantial spatial variability in snowfall this may affect results.

We chose stations near large population centers as we thought the results on snowfall had interesting and important societal implications. We did perform the analysis on more stations than we included in the manuscript and the results are consistent across all the stations in the Eastern US.

2. I do grasp that in order to improve sample size, a broader net of what is AA needs to be cast. This said, why is 1990 the start date? It seems like a focus just on the last decade (since 2007) would be insightful, or even a broader period back to the late 1990s.

We do agree that a start date of 2007 would be interesting and useful. However, a start date since 1990 is also interesting and useful as the review paper on the subject AA (Cohen et al., 2014) is identified as the starting point for the era of AA. And as the reviewer correctly points out, starting in 1990 improves the sample size.

3. Further, the authors note the trends are most substantial during the typical time of SSW (line 184); did the authors attempt to partition the relationship out by time of the season? That is, to focus on the mid-to-late winter, when the Arctic – mid-latitude relationships seem most robust?

This is an interesting idea that the reviewer raises that we did not consider earlier. We did as the reviewer suggested and the results were mixed. However, the relationship between PCT in the stratosphere and the AWSSI improved for late winter in the period of AA. We did not include the figure but included some additional text.

Same as Supplementary Figure 3 for AA period (bottom plot) but only for late winter.

On Limitations:

4. The AWSSI is an interesting and useful index. One issue of concern is that the way it was defined was with a mid-latitude continental climate of the midwest in mind, and hence by its nature it will be most responsive in this region. It is thus in one sense unsurprising to see the strongest relationships in the region surrounding where it was defined. More caution should be used in interpreting the results – in ‘marginal’ places like Atlanta and Seattle, single events are going to have substantial influence on overall values, more so than in the more continental locations. A little more contextualization on what the AWSSI is, and what it conveys and its limitations, would help understand the results better.

We agree with the reviewer and have included additional text contextualizing the AWSSI when it is first introduced.

Other comments:

5. Line 32-34: There are a lot of studies that do not show widespread cooling on NH continents in recent decades; in Figure 9c in the paper itself, something approaching ‘widespread’ could only be claimed for Asia. It seems to be much more so a lack of increase, and much greater year-to-year variability. This sentence should be modified to reflect this.

The sentence is only citing previous work (and faithfully represented that work); still, the sentence has been modified to reflect the concern of the reviewer.

6. Paragraph in lines 61-67... it might be best to put the last sentence first, to make it clearer

that the three variables are not being independently assessed, but rather it's the relationship among them that you're examining.

We edited the text as suggested by the reviewer.

7. Line 118 – change it's to it is.

Changed.

8. Line 199-200: The authors state that the 'relationship is fairly consistent throughout the troposphere, for both sub-periods...' In Figure S2, the two periods look vastly different, with an overall weaker relationship in the AA period. What am I missing?

We agree that the relationship is stronger in the earlier period, however, qualitatively the relationship is the same—severe winter weather is more common when PCHs are elevated. We edited the text to reflect the concerns of the reviewer.

9. Line 211 – should be Figure 8, not Figure 3. With regard to this figure and discussion, are the differences statistically significant?

The reviewer is correct and the figure has been corrected. For the revised manuscript, we computed statistical significance. We used a Wilcoxon non-parametric test for statistical significance in snowfall return, given the discontinuous nature of snowfall events and no reason to believe variances would be equal between 1950-1989 and 1990-2016 partitions. Using this metric, the snow fall returns were statistically significant at $p < 0.5$ for Seattle, Helena, Salt Lake City, Des Moines, Atlanta, Duluth, Washington and Boston (Blue Hill).

10. Lines 367-371. Authors should mention the subset of months used here within each calendar year. Also, for anomalies, are these anomalies de-seasonalized too? If so, how?

The study was conducted for the three winter months December, February and January, which we now include in the text. We did remove seasonality from the variables analyzed.

11. Figures: Blue Hill line points to Philadelphia, and there is no Philadelphia line.

The lines from the plots point to the correct stations but it is confusing because the plot for Boston covers the line connecting Philadelphia to its plot as well as the line from Boston to its plot. The plots have been changed in the revised manuscript and is no longer an issue.

12. Figure 5. I don't agree with the figure caption "by at least two weeks", since the authors explore 5-14 days here – thus there is no evidence of the "at least" part.

We removed "at least."

13. Figure 7. I'd suggest making the eastern city lines one range of hues, and the western ones another, and also organize the legend by east vs. west, to facilitate understanding

We thank the reviewer for his suggestion, which we adopted.

14. Figure 9c. Was statistical significance testing done, and nothing was significant? Or was it not done?

We did not compute statistical significance in the original submission but have done so in the revised manuscript.

Reviewer #3 (Remarks to the Author):

We are very grateful to this reviewer for these constructive comments and suggestions that have improved the manuscript significantly.

Their results will be a valuable contribution to the scientific discussion and also interesting for the general reader of Nature Communication.

Following, I have listed three main points regarding method, figures, and general understanding that I would like to see addressed and some minor comments are given below.

We thank the reviewer for their encouraging words.

(i) Method

I am a little bit concerned about the high and frequent significance of the presented results and the underlying statistical analyses. For example, daily data within one winter season will be highly auto-correlated. The applied t-test, however, requires independent variables. Has this been taken into account when estimating significance? This point is crucial for most of the presented results.

As I understand, significance in Figure 1 (and others) has been calculated by comparing negative and positive anomaly bins. But how exactly has this been done? Did you compare all AWSSI values of a given pressure level associated with negative PCH anomalies with all respective AWSSI values associated with positive PCH anomalies? For Figure 1 PCH and PCT have been standardized. But what about AWSSI? I would expect that this time series needs to be at least detrended. In the abstract you mention a robust and linear relationship between PCH/PCT and AWSSI. Do you extract this from Figure 1? If so I would like to see more quantitative evidence for this, particularly, because from the figure itself a linear relationship between PCT and AWSSI seems to be absent for many cities.

We detrended the data and accounted for autocorrelation. There was strong autocorrelation and when we accounted for autocorrelation in our statistical significance testing, the values dropped but still remain highly significant. We did compute the linear trends in the daily change in the AWSSI, the PCH and PCT and the trends were quite small ($<.001$) and mixed. We reproduced Figure 1 and Supplementary Figure 2 with detrended data and could see little to no visible difference in the results (see attached figure). We did not change those figures but in Supplementary Table 1 the values we list are those tested for statistical significance with the detrended data and accounted for autocorrelation.

Same as Figure 1 for PCH (top) with detrended data.

Figure 6 and figure 9 should be repeated using field significance testing (see Wilks, BAMS, 2016).

We have now accounted for field significance, which is mentioned in the figure captions for Figures 6 and 9 and in the Methods. For the analysis of 500 hPa PCH and PCT anomalies versus 2-m temperature anomalies (Figure 6), we verified field significance using the technique outlined in Livezey and Chen (1983). For these analyses, significance at $p < 0.05$ required at least 8.6% (PCT) to 9.4% (PCH) of the area meet criteria, which was exceeded for both cases.

We also ensured field significance for the 1000 hPa NAM and PCT analyses (Figure 9a and 9b), finding the total significant area must exceed 8.3% (NAM) and 9.2% (PCT). Similar verification for the 850 hPa Barents-Kara Sea (Figure 9d) confirmed significance for areal coverage exceeding 6.4%. For the surface temperature difference between the NAM and PCT, over 90% of the Northern Hemisphere was found to be significant at the 95% confidence level.

(ii) Figures.

In my print out it was basically impossible to read the text and numbers in those figures showing the US map. I think it would be enough to show the map with the station locations once in the beginning and then use the same ordering of stations for all figures. Please add units to all x and y axes.

We revised the figures to improve clarity.

(iii) General understanding.

I found it confusing that on p. 4 the authors refer to "warm Arctic" when in fact they talk about positive PCH anomalies. This is particularly confusing, because in the lower panel of Figure 1

they do show Arctic temperatures. This misunderstanding is partly solved in the paragraph from line 95-101 which I think is very important and should be explained more (including refs), particular, because this may not be clear to the general audience of Nature Communications. Having said this, it would improve readability if the authors clarified from the beginning why they look at PCH and PCT (and not NAM for example).

We moved up the discussion on lines 95-101, expanded the discussion and included a reference.

To me it looks like the correlation between PCH and AWSSI is much stronger in the pre-AA era (Supplementary Figure 2). This should be discussed in more detail since the influence of AA is an important part of the study.

We included more text on the relationship between PCH and the AWSSI and the difference before and after AA.

For a quantitative analysis – which the authors claim distinguishes their analysis from others – I would actually expect a sentence like “for positive PCH anomalies X standard deviations above the climatological mean we find a Y% increase in severe winter extremes” or similar.

We liked the reviewer’s suggestion. We tried to quantify the change in extreme winter weather as suggested by the reviewer. We included the following text: “We found in the lower stratosphere to mid troposphere (70 to 500 hPa), that a positive PCH anomaly of two standard deviations or greater is associated with two to four times more likely severe winter extremes. These extremes were typically on the order of two to six standard deviations based on AWSSI. This effect was most apparent in those stations in the northeastern and upper midwestern US.”

Minor

- *l. 51: I am not perfectly familiar with the literature but somehow I would be surprised if there were no other studies which are not constrained by these limitations.*

We are not familiar with other studies that are more comprehensive.

- *l. 57: I would add which pressure levels have been used since this is important for PHC*

We included the pressure levels.

- *l. 59: Poleward of 60°N*

For this study, we chose 65°N because of the PCT to better differentiate between the warm Arctic and cold continents.

- *l. 78: probably correct, but just from the figures peak AWSSI could in many cases be somewhere between 1.5 and 2.5 PCH anomaly*

We edited the text to be more precise.

- *l. 111: here you only show the link to U.S. mid-latitudes*

We edited the text.

- *l. 112: weekend polar vortex in the troposphere or stratosphere or both?*

We included “stratospheric” to be more precise.

- *l. 119: what do you mean by amplified flow?*

Larger Rossby waves, now included.

- *l. 185 weather has increased in eastern US.*

“Eastern US” has been included in the sentence to be clearer.

- *l. 211: Figure 3 should be Figure 8*

Corrected, thank you.

- *l. 217: This statement is too strong in my opinion, because the presented analysis is simply based on two different time periods*

We weakened the language.

- *l. 341: [...] two Arctic-only indicators and a new index [...]*

Sentence has been changed.

- *l. 412: I did not quite understand how exactly you compute the return times. Also I found the example slightly confusing because return times of “winters” are based on 365 days per year and “events” are based on 90 days per year. Here, again, auto-correlation may be important.*

We edited the text to make this clearer. We only considered snowfall that occurred for the three months of December through February but included all 365 days of the year when computing return times. So for example A 3-inch snowfall may occur four times in a single winter, about 90 days. An 18-inch snowfall may occur roughly every two winters, or 730 days. We talked through several strategies to describe this effectively and the return time as

plotted was the outcome. It is hard to see how autocorrelation is a factor with snowfall as it is so episodic. We computed statistical significance but did not account for auto-correlation.

- *l. 654: There are many examples where $p \geq 0.01$*

We recomputed the significance with more stringent statistics.

- *SI l. 1: difference*

Corrected.

- *SI l. 21: Fig.1 should be Fig. 2*

Corrected.

Reference:

Livezey, R. E. & Chen, W. Y. Statistical field significance and its determination by Monte Carlo Techniques. *Mon. Wea. Review.* **111**, 46–59 (1983).

Reviewers' comments:

Reviewer #1 (Remarks to the Author):

I thank the authors for addressing my comments. Their responses were satisfactory to convince me. Now I support the paper's publication. I note below a few very minor comment.

I now understand the meaning of "daily change in AWSSI". Then, a word of "change" might be misleading. Should authors use "daily score" instead?

Additional analyses using NAM index support that Arctic metrics PCH/PCT better represent the Arctic warming and relate to winter severity over the U.S. It is interesting and gives a new insight for the Arctic amplification. Readers might be interested in if this were applicable over NH mid-latitude other than U.S. I think it is beyond of a scope of this study. Then, I have a concern as that title of this study is a bit exaggerated. I recommend that title would be "*** in northern North America" or "*** in U.S.".

Reviewer #2 (Remarks to the Author):

I think the revision has addressed all of my concerns well, and will make a good contribution to the literature. I only have one minor issue: while I agree that the figures are now much clearer, it seems as though there are now no figures showing the study locations. There should be a map showing the AWSSI stations somewhere in the main document for reference.

Reviewer #3 (Remarks to the Author):

I much appreciate the author's effort to significantly improve the paper. I think it is in very good shape now and most of my concerns have been well addressed. Below you find my comments.

There is one very critical point in the applied method that has not convinced me so far. That is how statistical significance has been calculated for Fig. 1 (and probably Fig.6 and Fig.9). I appreciate that the authors detrended the data and accounted for autocorrelation. Unfortunately, they did not explain how this was done. And I still do not see how they were able to apply a Student's t-test to the data.

Autocorrelation may actually also be a problem for significance testing in Fig. 6 and Fig. 9. I acknowledge that the authors calculated field significance but this will only produce meaningful results if the calculation of each grid cell's p-value was done properly by accounting for autocorrelation.

Moreover, it would be interesting to know which regions were included for the calculation of field significance. Because the statement drawn from Fig. 6 for example is about warm Arctic cold continents I would highly recommend to only use respective grid cells and exclude for example ocean regions. Also it would be a very quick exercise and I would very much like to see these two figures with "local" field significance as described in [Wilks, 2016].

Minor comments

- L207: original sentence without ", which exhibits" should be correct
- L391: is 60°N correct?
- L393: Does "aggregated into daily values" mean that you calculate mean and variance for each calendar day of the winter season?

- Fig.1: in Title: 1950-2016 (see also Fig. 2 and Fig. 3)
- Fig.1: Second panel of Fig.1 is missing
- L690: is 60°N correct?
- L723: [-0.5, -0.3]

Wilks, D. S. (2016), "The Stippling Shows Statistically Significant Grid Points": How Research Results are Routinely Overstated and Overinterpreted, and What to Do about It, *Bull. Am. Meteorol. Soc.*, 97(12), 2263–2273, doi:10.1175/BAMS-D-15-00267.1.

Response to Reviewers for “Warm Arctic episodes linked with extreme winter weather in Northern Hemisphere mid-latitudes” by Cohen et al.

Reviewer #1

I thank the authors for addressing my comments. Their responses were satisfactory to convince me. Now I support the paper's publication. I note below a few very minor comment.

We are grateful to the reviewer for the encouraging words.

I now understand the meaning of “daily change in AWSSI”. Then, a word of “change” might be misleading. Should authors use “daily score” instead?

We changed “the daily change in AWSSI” to “the daily increment to the seasonal value of the AWSSI.”

Additional analyses using NAM index support that Arctic metrics PCH/PCT better represent the Arctic warming and relate to winter severity over the U.S. It is interesting and gives a new insight for the Arctic amplification. Readers might be interested in if this were applicable over NH mid-latitude other than U.S. I think it is beyond of a scope of this study. Then, I have a concern as that title of this study is a bit exaggerated. I recommend that title would be “*** in northern North America” or “*** in U.S.”.

We have changed the title as suggested by the reviewer.

Reviewer #2 (Remarks to the Author):

I think the revision has addressed all of my concerns well, and will make a good contribution to the literature.

We are very grateful to Dr. Sheridan for this encouraging comment.

I only have one minor issue: while I agree that the figures are now much clearer, it seems as though there are now no figures showing the study locations. There should be a map showing the AWSSI stations somewhere in the main document for reference.

All the stations used in the study are displayed in a map of the United States in Figure 1.

Reviewer #3 (Remarks to the Author):

I much appreciate the author's effort to significantly improve the paper. I think it is in very good shape now and most of my concerns have been well addressed. Below you find my comments.

We are grateful to the reviewer for these additional encouraging comments. Also, the suggested edits have helped to further improve the manuscript.

There is one very critical point in the applied method that has not convinced me so far. That is how statistical significance has been calculated for Fig. 1 (and probably Fig.6 and Fig.9). I appreciate that the authors detrended the data and accounted for autocorrelation. Unfortunately, they did not explain how this was done. And I still do not see how they were able to apply a Student's t-test to the data.

We now included in the Methods section the following explanation of the statistical testing. The data used in Figures 1, 6 and 9 were detrended by removing the linear trend in the time series prior to binning the data. The Student's t-test was applied to the bins pairwise as described in the text, adjusting downward the degrees of freedom based on the autocorrelation in the timeseries.

Autocorrelation may actually also be a problem for significance testing in Fig. 6 and Fig. 9. I acknowledge that the authors calculated field significance but this will only produce meaningful results if the calculation of each grid cell's p-value was done properly by accounting for autocorrelation.

Moreover, it would be interesting to know which regions were included for the calculation of field significance. Because the statement drawn from Fig. 6 for example is about warm Arctic cold continents I would highly recommend to only use respective grid cells and exclude for example ocean regions. Also it would be a very quick exercise and I would very much like to see these two figures with "local" field significance as described in [Wilks, 2016].

We computed statistical significance as suggested by the reviewer in Figures 6 and 9 now also accounting for local field significance as described in Wilks. The new figures are included in the revised manuscript. It did result in some changes in the statistical significance (mostly over the oceans) but none that changed the conclusions in the manuscript. We agree with the reviewer that we are mostly focused on the "warm Arctic/cold continents" pattern and therefore we applied a mask to all ocean grid points south of 60°N.

Minor comments

- L207: original sentence without ", which exhibits" should be correct

We deleted ", which exhibits."

- L391: is 60°N correct?

The reviewer is correct, it should be 65°N.

- *L393: Does “aggregated into daily values” mean that you calculate mean and variance for each calendar day of the winter season?*

Yes but the sentence includes some redundant language and has been deleted.

- *Fig.1: in Title: 1950-2016 (see also Fig. 2 and Fig. 3)*

We included the dash in the title.

- *Fig.1: Second panel of Fig.1 is missing*

Thank you! Second panel now included.

- *L690: is 60°N correct?*

The reviewer is correct, it should be 65°N.

- *L723: [-0.5, -0.3]*

We removed the redundant negative sign.

Reviewers' Comments:

Reviewer #3:

Remarks to the Author:

I am very happy with the changes made and have no further comments.